

# Conformal line defects at finite temperature

Julien Barrat[1⋆], Bartomeu Fiol[2†], Enrico Marchetto[3‡],
Alessio Miscioscia[1∘] and Elli Pomoni[1§]

**1** Deutsches Elektronen-Synchrotron DESY, Notkestr. 85, 22607 Hamburg, Germany
**2** Departament de Física Quàntica i Astrofísica i Institut de Ciències del Cosmos,
Universitat de Barcelona, Martí i Franquès 1, 08028 Barcelona, Catalonia, Spain
**3** Mathematical Institute, University of Oxford, Andrew Wiles Building,
Woodstock Road, Oxford, OX2 6GG, U.K.

⋆ julien.barrat@desy.de , † bfiol@ub.edu , ‡ marchetto@maths.ox.ac.uk ,
∘ alessio.miscioscia@desy.de , § elli.pomoni@desy.de

## Abstract

We study conformal field theories at finite temperature in the presence of a temporal conformal line defect, wrapping the thermal circle, akin to a Polyakov loop in gauge theories. Although several symmetries of the conformal group are broken, the model can still be highly constrained from its features at zero temperature. In this work we show that the defect and bulk one and two-point correlators can be written as functions of zero-temperature data and thermal one-point functions (defect and bulk). The defect one-point functions are new data and they are induced by thermal effects of the bulk. For this new set of data we derive novel sum rules and establish a bootstrap problem for the thermal defect one-point functions from the KMS condition. We also comment on the behaviour of operators with large scaling dimensions. Additionally, we relate the free energy and entropy density to the OPE data through the one-point function of the stress-energy tensor. Our formalism is validated through analytical computations in generalized free scalar field theory, and we present new predictions for the $O(N)$ model with a magnetic impurity in the $\varepsilon$-expansion and the large $N$ limit.

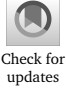

# 1 Introduction

Conformal Field Theory (CFT) finds applications across various areas of physics, ranging from condensed-matter systems at the critical point to quantum gravity via the AdS/CFT correspondence. The study of defects within CFTs is also fundamental since these describe a wide range of physical phenomena, such as magnetic impurities in materials with emergent Lorentz symmetry (see, e.g., [1,2] and references therein), or the radiation of moving quarks in high-energy physics [3–5]. A notable class of defects, called conformal defects, breaks the conformal symmetry in a controlled manner, with the residual symmetry imposing significant constraints on observables. This has led to the development of the defect bootstrap program [6–10]. However, much less is known about these models at finite temperature. In this case, a scale (the inverse temperature $\beta$) is introduced by considering the defect CFT on the geometry $S_\beta^1 \times \mathbb{R}^{d-1}$.

This paper focuses on the scenario of a temporal line defect, wrapping the thermal circle $S_\beta^1$, inspired by its relevance to the confinement problem and the AdS/CFT correspondence. Indeed, in gauge theories, the thermal analogue of the Wilson loop, known as the Polyakov loop, acts as an order parameter for the confinement/deconfinement phase transition [11]. Although confinement does not occur in a CFT, studying this setup provides a context endowed with analytical tools. Additionally, thermal CFTs are holographically dual to black holes in AdS, making the study of defects in this framework both fascinating and crucial [12]. In condensed-matter physics, line defects wrapping the thermal circle describe point-like impurities, which can be experimentally probed [13, 14].

Although the conformality of the bulk theory is broken by both thermal effects and the presence of the defect, many local properties of the zero temperature CFT remain useful for characterizing the system. Thermal correlation functions (without a defect) can be analyzed using the Operator Product Expansion (OPE), albeit with a finite radius of convergence and additional OPE data in the form of thermal one-point functions. At zero temperature, the associativity of the OPE leads to highly constraining consistency conditions. At finite temperature, an equivalent condition was identified as the *Kubo-Martin-Schwinger* (KMS) condition [15,16]. This approach is central to the bootstrap methods for thermal CFTs, initiated in the seminal work [16] and further developed in subsequent studies [17–21]. Similarly, the bootstrap approach to defect CFTs at zero temperature introduces new defect OPE data. Specifically, the presence of a conformal line defect results in a *one-dimensional* CFT living on the defect. Such setups have received significant attention recently, with bootstrap methods yielding a wealth of novel results [22–33].

This work aims to construct a bootstrap framework for a thermal defect CFT, specifically focusing on a line defect wrapping the thermal circle. We find that the *minimal* set of OPE data required to fully characterize a specific model is:

1. *Zero temperature CFT data*: the scaling dimensions $\Delta_{\mathcal{O}}$ and the structure constants $\mu_{ijk}$ of the bulk CFT.

2. *Finite temperature data*: the thermal one-point functions of bulk operators $b_{\mathcal{O}}$ (in the absence of the defect).

3. *Zero temperature defect data*: the scaling dimensions $\widehat{\Delta}_{\widehat{\mathcal{O}}}$ and the structure constants $\widehat{\mu}_{ijk}$ of the defect CFT, as well as the bulk-to-defect coefficients $\lambda_{\mathcal{O}\widehat{\mathcal{O}}}$.

4. *Thermal + defect data*: the new data consists of the one-point functions of defect operators $\widehat{b}_{\widehat{\mathcal{O}}}$.

The CFT and defect data are inherited from the zero temperature theory and remain unaltered by thermal effects. Thermal one-point functions of bulk operators are known only for a few specific cases, but they are the subject of the currently developing finite temperature bootstrap program. In our approach, these three classes of data will be considered as *input*. In this work, the one-point functions of the one-dimensional theory on the defect are the new data we are interested to study. As we shall explain, these defect one-point functions are a consequence of the thermal excitations of the bulk. Therefore the $1d$ theory living on the defect is more than just a thermal CFT.

The paper is structured as follows:

* in Section 2, we start with an analysis of the symmetries of the system, and in particular we show how correlation functions are constrained by the OPE data listed above. We provide an explicit expansion for one-point functions of bulk operators and present an inversion formula to extract the defect one-point functions,

* in Section 3, we set up a bootstrap problem for the new data $\widehat{b}_{\widehat{\mathcal{O}}}$, by deriving explicit sum rules from the KMS condition imposed on a two-point function of identical bulk operators in the presence of the defect. We discuss some implications and how the behaviour of heavy operators can be predicted,

* in Section 4, we study the thermodynamics of the system in $d > 2$. In particular, we relate the free energy and the entropy density of the system to the defect OPE of the stress-energy tensor. We also comment on the defect entropy and the free energy of a moving quark in gauge theories,

⋆ in Section 5, we present applications to specific models. We test our equations in the case of Generalized Free (scalar) Field (GFF) theory, and we obtain new results for the case of the $d$-dimensional O($N$) model, both in the contexts of the $\varepsilon$-expansion and at large $N$.

Finally, in Section 6, we conclude with a discussion of our results and a presentation of potential future prospects. In Appendix A, we present the (broken) Ward identities for the conformal group in an explicit form, while in Appendix B, we comment on the real-time formalism, which could provide a different approach to the same problem. In Appendix C we review the defect thermodynamics of two-dimensional CFTs and finally in Appendix D we present a technical aspect regarding the bulk to defect OPE of the stress-energy tensor which is important for the finiteness of the entropy in the zero temperature limit.

## 2 Defect CFT at finite temperature

In this section, we discuss the symmetries of a CFT with a defect at finite temperature. We present possible configurations and analyze which symmetries are broken or preserved when the defect wraps the thermal circle. We then examine the correlation functions associated with the thermal defect CFT and identify the OPE data required for a complete characterization. We conclude by presenting an inversion formula for the defect one-point functions.

### 2.1 Line defects on the thermal manifold

#### 2.1.1 Configurations

Our starting point is a Euclidean CFT at zero temperature, i.e. a CFT on the flat manifold $\mathbb{R}^d$. We label the representations of the conformal group by the scaling dimensions $\Delta$ and the spin $J$. A *defect CFT* can be defined by inserting an extended operator of codimension $q$ (called a *conformal defect*) that preserves part of the original symmetry. Specifically, for a conformal line defect, the codimension is $q = d - 1$ and the symmetry is broken in the following way

$$\text{SO}(d+1,1) \longrightarrow \text{SO}(2,1) \times \text{SO}(d-1). \tag{1}$$

The SO(2, 1) group can be interpreted as the one-dimensional conformal symmetry preserved along the defect, while SO($d-1$) corresponds to the unbroken rotations around the defect. From the $1d$ perspective, the latter can be viewed as a *global* symmetry [34,35]. Defect representations are then labeled by the quantum numbers $\widehat{\Delta}$, the $1d$ scaling dimension, and $s$, the transverse spin, associated to the group SO($d-1$).

Our goal is to study this system at finite temperature. More precisely, we consider the defect CFT not on flat space, but on the *thermal manifold*

$$\mathcal{M}_\beta = S^1_\beta \times \mathbb{R}^{d-1}, \tag{2}$$

where the original time direction is compactified along a circle of length $\beta = 1/T$, with $T$ being the temperature. In the absence of a defect, this compactification explicitly breaks the conformal symmetry of the bulk theory [16, 20]. Contrary to the zero temperature case, the orientation of the line defect plays a significant role in how the symmetries are broken or preserved. We can consider *three* different systems:

(a) The line defect wraps the thermal circle.

(b) The line defect is placed along one of the (not compactified) space directions.

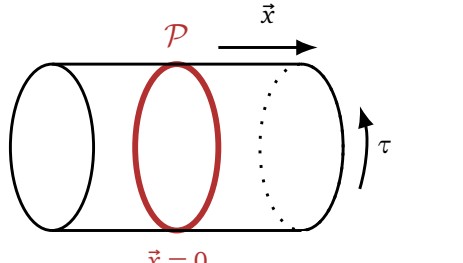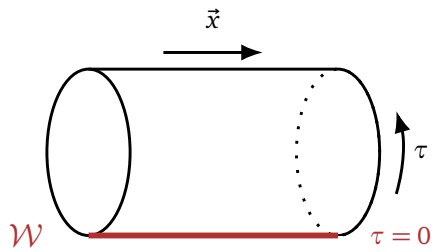

Figure 1: The left figure shows a temporal line defect wrapping the thermal circle at the spatial position $\vec{x} = 0$. This configuration corresponds to case (a) in the main text, and is the focus of this paper. On the right side, the setup (b) is illustrated, for which the spatial line defect is placed at $\tau = 0$ and extends in a spatial direction.

(c) The line defect is placed in a direction that includes both a component in time and space. In two dimensions, this defect can for instance take the shape of a *spiral* wrapping the thermal cylinder.

Graphical representations of cases (a) and (b) are shown in Fig. 1. At zero temperature, all configurations are equivalent in Euclidean space. However, at finite temperature, these different setups describe *distinct physical systems*. For instance, in setup (a), the defect preserves translations along the time direction and preserves a symmetry group $SO(d-1)$. In contrast, in setup (b), only rotations orthogonal to the defect remain unbroken, reducing the symmetry group to $SO(d-2)$.

Although all configurations have significant physical applications, this paper focuses on type (a) which maximizes the residual conformal symmetry of the defect CFT. Importantly, in the context of gauge theories, this configuration corresponds to *Polyakov loops*, whose one-point function acts as the order parameter for the confinement/deconfinement transition.

### 2.1.2 Broken and unbroken symmetries

Let us summarize the effect of turning on the temperature on the symmetries of the conformal group for the case where the defect wraps the thermal circle. Building upon the discussion of the previous Section, we have:

* **Translations**: Time translations around the thermal circle are unbroken, while spatial translations are broken by the presence of the defect.

* **Rotations**: Spatial rotations are unbroken, while boosts are broken by both the defect and finite temperature effects.

* **Dilatations and SCTs**: Dilatations and special conformal transformations (SCTs) are broken only by finite temperature effects.

* **Global symmetries**: Any global symmetry is preserved at finite temperature, but can be broken by the defect. The potential breaking of external symmetries by the defect depends on the details of the theory.

This set of symmetries defines the *thermal defect CFT* that we study in the rest of this work. It is possible to derive (broken) Ward identities in the spirit of [20]. This results in non-trivial constraints on the correlation functions of the theory. These (broken) Ward identities are derived and presented in Appendix A.

## 2.2 Correlation functions

We now turn our attention to the implications of these symmetries: our aim is to provide the *OPE data* necessary to (in principle) fully solve the system through repeated use of the bulk and defect OPEs.

### 2.2.1 Bulk and defect OPEs

One crucial aspect of this setup is that the thermal geometry described by (2) is *conformally flat*.[1] This implies that *local* properties of the zero temperature theory are preserved when the time dimension is compactified. In other words, the scaling dimensions of the operators remain unaffected, and products of two (or more) operators can be expanded using the same OPE as at zero temperature. For instance, the *bulk OPE* for two identical scalar operators $\phi$ of dimension $\Delta_\phi$ is given by

$$\phi(x_1)\phi(x_2) = \frac{1}{x_{12}^{2\Delta_\phi}} \sum_{\mathcal{O}\in\phi\times\phi} \mu_{\phi\phi\mathcal{O}} |x_{12}|^{\Delta-J} x_{12\,\mu_1} \cdots x_{12\,\mu_J} \mathcal{O}^{\mu_1\cdots\mu_J}(x_2), \tag{3}$$

where the conformal data (i.e., the scaling dimensions $\Delta$ and the structure constants $\mu_{\phi\phi\mathcal{O}}$) appearing in this equation are zero temperature data. Before proceeding let us notice that the OPE convergence, discussed in similar CFT contexts in [36–38], is now limited to a *finite* region. Concretely, the convergent region of the bulk OPE (3) in the presence of the defect is given by

$$\tau_{12}^2 + \vec{x}_{12}^2 \leq \min\left\{\beta^2, |\vec{x}_1|^2, |\vec{x}_2|^2\right\}, \tag{4}$$

with $|\vec{x}_i|$ the transverse position of the two operators with respect to the defect.

In defect CFT, it is well-known that bulk operators can be expanded in terms of the defect operators introduced in Section 2.1.1, by bringing the bulk operator close to the defect. For a scalar operator $\phi$, this results in the *defect OPE*

$$\phi(\tau,\vec{x}) = \frac{1}{|\vec{x}|^\Delta} \sum_{\widehat{\mathcal{O}}} \lambda_{\phi\widehat{\mathcal{O}}} |\vec{x}|^{\widehat{\Delta}-s} x_{i_1} \cdots x_{i_s} \widehat{\mathcal{O}}^{i_1\cdots i_s}(\tau), \tag{5}$$

where $|\vec{x}|$ is the transverse distance between the operator and the defect. The OPE written above is expected to be convergent in the region $|\vec{x}| < \beta$, unless other operators are closer to the defect. Once again, the OPE data appearing in this equation corresponds to the zero temperature case. Here, the coefficients $\lambda_{\phi\widehat{\mathcal{O}}}$ describe bulk-defect two-point functions, which are fixed kinematically at $T = 0$ [7].

### 2.2.2 Bulk and defect one-point functions

Non-local properties of the theory can however be affected by the compactification of the time dimension. Since dilatations are not preserved, one-point functions of local operators (without defects) can in general be non-zero. Translational invariance along the thermal circle fixes them to be constant and depend on a single coefficient [16,20]

$$\langle \mathcal{O}_\Delta^{\mu_1\cdots\mu_J} \rangle_\beta = \frac{b_\mathcal{O}}{\beta^\Delta} (e^{\mu_1} \cdots e^{\mu_J} - \text{traces}), \tag{6}$$

where $e^\mu$ is non-zero only when placed in the time direction. We refer to these correlators as *bulk* thermal one-point functions.

---

[1]A manifold is *conformally flat* if for each point there is a neighborhood that can be map to an open subset of the flat space via a conformal transformation. This does not imply the existence of a conformal map between the full thermal manifold and the flat space: this only happens in $d = 2$.

In order to understand the OPE data needed to characterize the thermal CFT in the presence of the defect, we study the first kinematically non-trivial correlator, i.e. the one-point function of a bulk operator in presence of the defect, that we denote by $\mathcal{P}$. At zero temperature, it is kinematically fixed, but since translations orthogonal to the defect are broken, the correlator is now given by

$$\langle \phi(\tau, \vec{x}) \mathcal{P} \rangle_\beta = \frac{F_\phi(z)}{|\vec{x}|^{\Delta_\phi}},\tag{7}$$

for a scalar operator $\phi$, with $z$ a dimensionless variable defined as

$$z = \frac{|\vec{x}|}{\beta}.\tag{8}$$

Using the defect OPE (5), we can rewrite (7) as

$$\langle \phi(\tau, \vec{x}) \mathcal{P} \rangle_\beta = \frac{1}{|\vec{x}|^{\Delta_\phi}} \sum_{\widehat{\mathcal{O}}} \lambda_{\phi \widehat{\mathcal{O}}} |\vec{x}|^{\widehat{\Delta}-s} x_{i_1} \cdots x_{i_s} \langle \widehat{\mathcal{O}}^{i_1 \cdots i_s}(\tau) \mathcal{P} \rangle_\beta,\tag{9}$$

which is convergent for $|\vec{x}| \leq \beta$. As mentioned in Section 2.1.1, the defect operators $\widehat{\mathcal{O}}$ are described by their quantum numbers $\widehat{\Delta}$ and $s$, corresponding respectively to the symmetry groups $SO(2,1)$ and $SO(d-1)$. Note that, at zero temperature, the correlator on the right-hand side vanishes for all operators except for the identity $\widehat{\mathbb{1}}$. From the invariance under time translations, we find that the defect one-point functions are given by

$$\langle \widehat{\mathcal{O}}_{\widehat{\Delta}}^{i_1 \cdots i_s} \rangle_\beta = \frac{\widehat{b}_{\widehat{\mathcal{O}}}}{\beta^{\widehat{\Delta}}} \Pi^{i_1 \cdots i_s},\tag{10}$$

with $\Pi^{i_1 \cdots i_s}$ a $SO(d-1)$-invariant tensor structure,[2] similarly to the bulk case. We refer to these correlators as *defect one-point functions* $\widehat{b}_{\widehat{\mathcal{O}}}$.

It should be emphasized at this point that these correlators do *not* correspond to one-dimensional thermal one-point functions. To appreciate this fact let us consider a $1d$ CFT compactified on $S_\beta^1$. In that case, the existence of a conformal map between the infinite line $\mathbb{R}$ and the circle $S_\beta^1$ and the absence of conformal anomalies immediately imply that the thermal one-point functions vanish. Naively, this seems to be in contradiction with (10) being non-trivial. However, *thermal effects in the bulk* induce additional one-point functions to the one dimensional defect theory living on $S_\beta^1$, as we will now explain.

As an intermediate step, we contract the $SO(d-1)$ tensor structure $\Pi^{i_1 \cdots i_s}$ with $x_{i_1} \cdots x_{i_s}$

$$x_{i_1} \cdots x_{i_s} \langle \widehat{\mathcal{O}}_{\widehat{\Delta}}^{i_1 \cdots i_s} \rangle_\beta = \widehat{b}_{\widehat{\mathcal{O}}} \frac{|\vec{x}|^s}{\beta^{\widehat{\Delta}}},\tag{11}$$

then (7) can be expressed as

$$F_\phi(z) = \sum_{\widehat{\mathcal{O}}} \lambda_{\phi \widehat{\mathcal{O}}} \widehat{b}_{\widehat{\mathcal{O}}} z^{\widehat{\Delta}}.\tag{12}$$

As a side comment, notice that operators with the same conformal dimension but different transverse spin appear with the same power of the dimensionless ratio $z$. In interacting theories, degeneracies are effectively rare, and considering several one-point functions of bulk operators can be used to lift the degeneracy.

---

[2]There is a normalization to choose in the definition of the tensor structure. In order to simplify some of the following expression we choose $x_{i_1} \cdots x_{i_s} \pi^{i_1 \cdots i_s} = |\vec{x}|^s$.

Table 1: Summary of the OPE data needed to characterize the thermal defect CFT, and how the coefficients are related to correlation functions at zero and finite temperature. Notice that the only two new sets of coefficients appearing at $T \neq 0$ are the thermal bulk one-point functions $b_{\mathcal{O}}$ and the defect one-point functions $\widehat{b}_{\widehat{\mathcal{O}}}$.

|  | bulk data | bulk-defect data | defect data |
|---|---|---|---|
| $T = 0$ | $\Delta_{\mathcal{O}} \sim \langle \mathcal{O}\mathcal{O} \rangle$ | $\lambda_{\mathcal{O}\widehat{\mathcal{O}}} \sim \langle \mathcal{O}\widehat{\mathcal{O}} \rangle$ | $\widehat{\Delta}_{\widehat{\mathcal{O}}} \sim \langle \widehat{\mathcal{O}}\widehat{\mathcal{O}} \rangle$ |
|  | $\mu_{ijk} \sim \langle \mathcal{O}_i\mathcal{O}_j\mathcal{O}_k \rangle$ |  | $\widehat{\mu}_{ijk} \sim \langle \widehat{\mathcal{O}}_i\widehat{\mathcal{O}}_j\widehat{\mathcal{O}}_k \rangle$ |
| $T = 1/\beta$ | $b_{\mathcal{O}} \sim \langle \mathcal{O} \rangle_{\beta}$ |  | $\widehat{b}_{\widehat{\mathcal{O}}} \sim \langle \widehat{\mathcal{O}} \rangle_{\beta}$ |

If now we consider the one-point function in the presence of a line defect(7), the function $F_{\phi}(z)$ necessarily obeys two special limits

$$F_{\phi}(z) \overset{z\to 0}{\sim} \lambda_{\phi\widehat{\mathbb{1}}}, \qquad F_{\phi}(z) \overset{z\to\infty}{\sim} b_{\phi}z^{\Delta}. \tag{13}$$

The first limit corresponds to the zero temperature case. Zero temperature one-point functions of the defect operators are zero apart from the identity.[3] The second limit instead corresponds to the high-temperature case. $b_{\phi}$ generically is non-zero: by looking at the equation (13), we deduce that the defect one-point functions $\widehat{b}_{\widehat{\mathcal{O}}}$ can be non-zero in general. Notice that the only physically meaningful quantity is the ratio $z$, hence the two limits can also be seen respectively as short and large distance from the line.

Keeping all the above in mind, any correlation function of this *thermal defect CFT* can (in principle) be decomposed by using the bulk and defect OPEs, and be expressed in terms of zero temperature data, namely the scaling dimensions and structure constants, to which we need to add the thermal one-point functions (without defect) and the defect one-point functions. For instance, two-point functions of defect operators $\widehat{\phi}_1$ and $\widehat{\phi}_2$ can be expressed in terms of the coefficients $\widehat{\mu}_{\widehat{\phi}_1\widehat{\phi}_2\widehat{\mathcal{O}}}$ and $\widehat{b}_{\widehat{\mathcal{O}}}$. This set of elementary OPE data is summarized in Table 1.[4] Let us comment that in principle the zero temperature CFT data suffices to compute any thermal correlation function since those are defined as correlators computed in the state defined by the density matrix $\rho = e^{-\beta H}$ (we review this formalism in Appendix B). Nonetheless, using this fact explicitly is very hard, even when the defect is not present [16]. On the contrary, we show in Section 3 that a bootstrap problem for the new data $\widehat{b}_{\widehat{\mathcal{O}}}$ is very similar to the case of the thermal CFT without defect.

### 2.2.3 An inversion formula for defect one-point functions

We have seen in the previous Section that defect one-point functions are the fundamental quantities to know, but also that they are induced by the bulk theory as thermal excitations. We now show how these coefficients can be determined through a computation in the bulk. Indeed, it is possible to write the OPE expansion of $F_{\phi}(z)$ (defined in (7)) as

$$F_{\phi}(z) = \int_{-\varepsilon-i\infty}^{-\varepsilon+i\infty} \mathrm{d}\Delta \, \frac{a(\Delta)}{2\pi i} z^{\Delta}, \tag{14}$$

---

[3]We implicitly use the normalization $\langle \widehat{\mathbb{1}} \rangle = \langle \widehat{\mathbb{1}} \rangle_{\beta} = 1$.

[4]At zero temperature, certain physical observables are related to the normalization of the two-point functions of defect operators. This is for instance the case of the Bremsstrahlung function, which is related to the two-point function of the displacement operator. This normalization can be thought of as a function of the bulk-to-defect OPE coefficients, and for this reason, we choose not to insert them in the table.

where the function $a(\Delta)$ has poles at the scaling dimensions of the physical operators, i.e. the residue is fixed to be

$$a(\Delta) \sim -\frac{\lambda_{\phi\widehat{\mathcal{O}}}\widehat{b}_{\widehat{\mathcal{O}}}}{\Delta - \widehat{\Delta}}, \tag{15}$$

where, as a technical requirement, $a(\Delta)$ is required not to grow exponentially in the positive $\Delta$ half (complex) plane. Here, $\varepsilon$ parametrizes the contour in the complex plane, which can be chosen arbitrarily as far as the integral converges. It is possible to perform a Mellin transform on (14) in order to obtain the function $a(\Delta)$

$$a(\Delta) = \int_0^1 \mathrm{d}z \, z^{-\Delta-1} \, F_\phi(z). \tag{16}$$

From the expression (15), we obtain the inversion formula for the OPE coefficients

$$\lambda_{\phi\widehat{\mathcal{O}}}\widehat{b}_{\widehat{\mathcal{O}}} = \mathrm{Res}_{\Delta\to\widehat{\Delta}} \int_0^1 \mathrm{d}z \, z^{-\Delta-1} \, F_\phi(z). \tag{17}$$

This equation provides a way to extract defect one-point functions given a scalar bulk one-point function and the zero temperature structure constants $\lambda_{\phi\widehat{\mathcal{O}}}$.

To conclude, note that the operators are identified by their scaling dimensions in (17). As mentioned in the previous Section, degeneracy in the transverse spin cannot be resolved using bulk one-point functions. It is however possible that two degenerate operators appear in two different bulk computations, i.e., with different zero temperature coefficients, and in this case the degeneracy can in principle be resolved.

## 3 Setting up a bootstrap problem

This Section is dedicated to formulating a bootstrap problem for the defect one-point functions at finite temperature. We present novel sum rules derived from the KMS condition, that impose strong constraints on the coefficients $\widehat{b}_{\widehat{\mathcal{O}}}$. With this in mind, we will also discuss the behaviour of heavy operators.

### 3.1 From the KMS condition to sum rules

Our starting point is the KMS condition for two identical scalar operators $\phi(\tau, \vec{x})$. In order to write an equation in which both sides can be expanded in OPEs we can additionally use $\tau \to -\tau$.[5] Concretely,

$$0 = \langle \phi(\beta/2+\tau, \vec{x}_1)\phi(0, \vec{x}_2)\mathcal{P}\rangle_\beta - \langle \phi(\beta/2-\tau, \vec{x}_1)\phi(0, \vec{x}_2)\mathcal{P}\rangle_\beta. \tag{18}$$

We describe in the following how to use the bulk and defect OPEs in order to derive sum rules for the one-point function coefficients $\widehat{b}_{\widehat{\mathcal{O}}}$.

The first step is to use the bulk OPE given in (3) on the two terms in (18). Note that, for this expansion to be convergent, we need to place the operators such that $|x_{12}| < \min(|\vec{x}_1|, |\vec{x}_2|) < \beta$. We obtain the following equation

$$\sum_{\mathcal{O}\in\phi\times\phi} \mu_{\phi\phi\mathcal{O}} \langle \mathcal{O}^{\mu_1\cdots\mu_J}(\vec{x}_2)\mathcal{P}\rangle_\beta |v_+|^{\Delta-2\Delta_\phi-J} v_{+\mu_1}\cdots v_{+\mu_J}$$
$$= \sum_{\mathcal{O}\in\phi\times\phi} \mu_{\phi\phi\mathcal{O}} \langle \mathcal{O}^{\mu_1\cdots\mu_J}(\vec{x}_2)\mathcal{P}\rangle_\beta |v_-|^{\Delta-2\Delta_\phi-J} v_{-\mu_1}\cdots v_{-\mu_J}, \tag{19}$$

---

[5]For parity invariant theories $\tau \to -\tau$ is a symmetry. In general we can compose with parity action on one of the spatial direction to preserve the same $\mathbb{Z}_2$ symmetry [16].

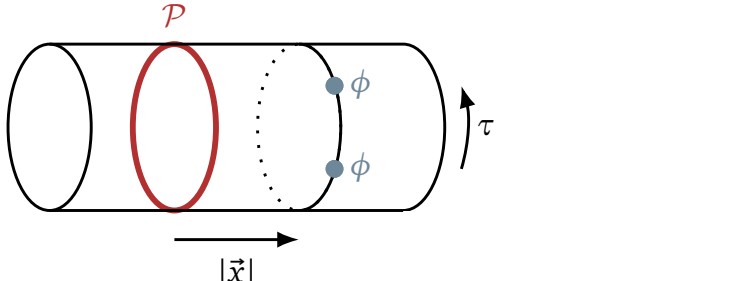
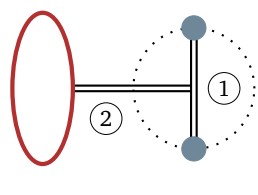

Figure 2: The configuration for the KMS condition in the collinear limit is depicted on the left. Here, the two local operators are located at a distance $|\vec{x}|$ from the Polyakov loop, and separated in the time direction by $\tau$. The figure on the right shows the order in which the OPEs must be taken in order to obtain the sum rules: (1) the bulk OPE $\phi \times \phi \to \sum \mathcal{O}$, (2) the defect OPE $\mathcal{O} \times \mathcal{P} \to \sum \widehat{\mathcal{O}}$. The dotted circle is here to emphasize the condition $\tau < |\vec{x}| < \beta$, necessary to be in the appropriate OPE regime.

where the vectors $v_\pm$ are defined through

$$v_\pm^\mu := (\beta/2 \pm \tau, \vec{x}_{12}), \tag{20}$$

which describe the distance between the two local operators. The expression simplifies considerably if the local operators are placed on the same thermal circle:[6]

$$v_\pm^\mu = \delta^{0\mu}\left(\frac{\beta}{2} \pm \tau\right). \tag{21}$$

The corresponding configuration is represented in Fig. 2. The resulting equation can then be expanded in $\tau$, and each term must vanish individually. This yields

$$0 = \sum_{\mathcal{O}} \mu_{\phi\phi\mathcal{O}} \langle \mathcal{O}^{0\cdots0}(\vec{x})\mathcal{P} \rangle_\beta \frac{\beta^\Delta}{2^\Delta} \binom{\Delta - 2\Delta_\phi}{n}, \tag{22}$$

with $n \in 2\mathbb{N} + 1$, as the terms with even powers cancel exactly.

The next step is to perform the defect OPE given in (5) for the operators $\mathcal{O}^{0\cdots0}(\vec{x})$. Note that the choice of placing the two operators at the same distance from the defect, namely the choice in equation (21), introduces two main simplifications: the operators effectively behave like scalars in this limit, and (5) can be used (see also (7)) without entering the problem of computing spinning blocks. Furthermore only primary operators contribute in the equation (22) because of translational invariance along the time direction.[7] Using the notation defined in Table 1 for the OPE coefficients, the following sum rules are obtained

$$0 = \sum_{\mathcal{O},\widehat{\mathcal{O}}} \mu_{\phi\phi\mathcal{O}} \lambda_{\mathcal{O}\widehat{\mathcal{O}}} \widehat{b}_{\widehat{\mathcal{O}}} \binom{\Delta - 2\Delta_\phi}{n} \frac{z^{\widehat{\Delta}-\Delta}}{2^\Delta}, \tag{23}$$

where $z$ is the cross-ratio defined in (8). This can be recast as

$$0 = \sum_{\Delta,\bar{\Delta}} \mu_{\phi\phi\mathcal{O}} \lambda_{\mathcal{O}\widehat{\mathcal{O}}} \widehat{b}_{\widehat{\mathcal{O}}} \binom{\Delta - 2\Delta_\phi}{n} \frac{z^{\bar{\Delta}}}{2^\Delta}, \quad \bar{\Delta} = \widehat{\Delta} - \Delta. \tag{24}$$

---

[6]In the conventions of, e.g. [8], this *collinear limit* can be understood as setting the cross-ratios equal to one another, i.e. $z = \bar{z}$. Equivalently, it corresponds to $\cos\phi = 0$ in the notation of [7].

[7]Observe that in the OPE between $\phi \times \phi$ (Equation (19)) not only bulk primary operators contribute. Nevertheless, once the two operators $\phi$ are located on the same thermal circle, all the descendants are built by acting with time derivatives. Therefore, in this configuration, only bulk primary operators contribute to the OPE because of time translational invariance.

A summary of the order in which the OPEs have been applied is shown in Fig. 2. These sum rules impose strong constraints on the coefficients $\widehat{b}_{\widehat{\mathcal{O}}}$, which can be studied using a variety of bootstrap techniques. For thermal bulk CFTs, analogous equations have been presented in [21]. Observe that in [21] a generalization of these sum rules, in the case where the two operators are not placed at the same spatial coordinates, is also given. In the current case, the presence of the defect complicates the spin structure, making the problem more difficult to handle. However the equations (24) are labeled by two parameters $\bar{\Delta}$ and $n$, while in the case with no defect the analogous equations are labeled by only one parameter, $n$. Therefore the set of equations (24) is expected to be more constraining. Furthermore, numerical techniques have been proposed to obtain predictions for the bulk one-point functions of the 3$d$ Ising model [19].[8]

A few comments, concerning the sum rules (24), are in order before we conclude this Section. First, it is important to notice that they correspond to a restricted set of constraints, since we have used the KMS condition (18) in the collinear limit (21). In Section 5, the set of equations (24) is checked explicitly in the case of GFF. Second, there is *a priori* no positivity condition on the OPE coefficients of (24). Note in particular that the binomial coefficients take *negative* values for specific parameters.

## 3.2 Heavy operators

The sum rules derived in the previous Section are a consequence of the KMS condition, and relate an infinite number of operators in different channels. In particular, (18) relates *light* operators to *heavy* ones. At zero temperature, this results into a relation between the lightest operator, the defect identity, and the heavy operators [39–41]. Similar relations also exist at finite temperature without defects [21]. In this Section, we show how to extend these relations to our setup with a line defect wrapping the thermal circle.

Let us consider again the collinear limit, in which the two scalar fields in the bulk are placed on the same thermal circle. In this case, we have

$$\langle \phi(\tau, \vec{x})\phi(0, \vec{x})\mathcal{P}\rangle_\beta = \sum_{\mathcal{O}, \widehat{\mathcal{O}}} \mu_{\mathcal{O}\phi\phi} \lambda_{\mathcal{O}\widehat{\mathcal{O}}} \widehat{b}_{\widehat{\mathcal{O}}} \frac{\tau^{\Delta - 2\Delta_\phi}}{|\vec{x}|^\Delta} z^{\widehat{\Delta}}. \tag{25}$$

Notice that the defect identity certainly contributes to the bulk OPE, as the two scalars are identical and the projection of the bulk identity to the defect is simply $\mathbb{1} \to \widehat{\mathbb{1}}$. Therefore, we can select the operators $\mathcal{O}$ (exchanged in the bulk channel) and $\widehat{\mathcal{O}}$ (exchanged in the defect channel) such that $\Delta = \widehat{\Delta}$ by picking the zeroth order in $|\vec{x}|$ in the two-point function. This results in

$$f_\beta(\tau) = \langle \phi(\tau, \vec{x})\phi(0, \vec{x})\mathcal{P}\rangle_\beta\big|_{|\vec{x}|^0} = \sum_{\Delta = \widehat{\Delta}} \mu_{\mathcal{O}\phi\phi} \lambda_{\mathcal{O}\widehat{\mathcal{O}}} \frac{\tau^{\Delta - 2\Delta_\phi}}{\beta^\Delta}.$$

The KMS condition (including parity $\tau \to -\tau$) now reads

$$f_\beta(\tau) = f_\beta(\beta - \tau). \tag{26}$$

We can define an OPE density specific to this problem

$$\rho(\widetilde{\Delta}) = \sum_\Delta a_\Delta \delta(\widetilde{\Delta} - \Delta), \tag{27}$$

---

[8]In this case, predictions are made by using the zero temperature CFT data as *input*. It would be interesting to understand if the converse is also true, i.e., whether finite temperature sum rules impose new constraints on the zero temperature data [15, 21].

where $a_\Delta = \mu_{\mathcal{O}\phi\phi}\lambda_{\mathcal{O}\widehat{\mathcal{O}}}$. In the limit $\tau \to \beta$, the KMS equation for $f_\beta(\tau)$ becomes

$$\int_0^\infty d\Delta\, \rho(\Delta)\frac{\tau^{\Delta-2\Delta_\phi}}{\beta^\Delta} \sim \frac{1}{(\beta-\tau)^{2\Delta_\phi}}\,, \tag{28}$$

with the right-hand side of the equation being the projection to the identity contribution. From the inverse Laplace transform, one concludes that

$$\rho(\Delta) \overset{\Delta\to\infty}{\sim} \frac{\Delta^{2\Delta_\phi-1}}{\Gamma(2\Delta_\phi)}\left[1 + \mathcal{O}\left(\frac{1}{\Delta}\right)\right]. \tag{29}$$

Note that this result can also be obtained by using a suitable Ansatz $\rho(\Delta) = A\Delta^\alpha$ for the OPE density.

Derivations of this type are common throughout the physics literature, as a consequence of the channel duality (see e.g. [40–42]). It should be pointed out that it is not *per se* mathematically precise, as it can happen that the OPE density oscillates between two polynomial functions for high scaling dimensions (see Fig. 1 in [39]). In practice, this can only happen in case of a fine tuning between the parameters $a_\Delta$.[9] This issue can be circumvented by either proving or assuming the positivity of the coefficients $a_\Delta$ at large $\Delta$. In this case, (29) follows as a consequence of *Tauberian theorems*, which were introduced in physics in [43], and found applications in various branches of theoretical physics [36, 44–48]. It would be interesting to make the derivation of (29) rigorous by proving the positivity of $a_\Delta$ at large $\Delta$.

Finally, (29) should formally be understood as an *average*, since, for instance, the OPE density cannot grow continuously in the case of a discrete spectrum. In fact, the correct statement is rather

$$\int_0^\Delta d\widetilde{\Delta}\, \rho(\widetilde{\Delta}) \overset{\Delta\to\infty}{\sim} \frac{\Delta^{2\Delta_\phi}}{\Gamma(2\Delta_\phi+1)}\left[1 + \mathcal{O}\left(\frac{1}{\Delta}\right)\right]. \tag{30}$$

# 4 Defect thermodynamics

We now discuss the thermodynamics of defects at finite temperature. We determine the free energy and the entropy density of the system in terms of OPE data, starting from the one-point function of the stress-energy tensor in the presence of a line defect. To conclude, we discuss the free energy of moving quarks in gauge theories.

## 4.1 Free energy density and entropy

Among all bulk operators, a special role is played by the stress-energy tensor. It can be related to standard thermodynamic quantities, such as the free energy and the entropy of the system. As a starting point, let us consider the generic constraints arising from the preserved symmetry under spatial rotations. At zero temperature, its one-point function in the presence of a conformal line defect is fixed up to a single function of the marginal couplings [34]. At finite temperature, the one-point function is fixed in terms of *two* functions of the dimensionless ratio $z$ defined in (8) and the marginal couplings. Employing $SO(d-1)$-invariance together with the tracelessness condition we can write the Ansatz for $d > 2$

$$\langle T^{ij}(\tau,\vec{x})\mathcal{P}\rangle_\beta = \frac{f_1(z)}{|\vec{x}|^d}\delta^{ij} - 2\frac{x^i x^j}{|\vec{x}|^{d+2}}f_2(z)\,, \tag{31}$$

$$\langle T^{00}(\tau,\vec{x})\mathcal{P}\rangle_\beta = -\frac{(d-1)f_1(z)-2f_2(z)}{|\vec{x}|^d}\,, \qquad \langle T^{i0}(\tau,\vec{x})\mathcal{P}\rangle_\beta = 0\,. \tag{32}$$

---

[9]The most standard example in the mathematical literature is discussed in (3.12) in [21]. We refer to this paper for more technical details.

The case $d = 2$ is discussed in Appendix C. There is no additional constraint arising from symmetry. However, expanding the stress-energy tensor in the OPE regime is possible. For instance, the time-time component yields the equation

$$-(d-1)f_1(z) + 2f_2(z) = \sum_{\widehat{\mathcal{O}}} \lambda_{T\widehat{\mathcal{O}}} \widehat{b}_{\widehat{\mathcal{O}}} z^{\widehat{\Delta}}. \tag{33}$$

It is important to notice that $f_1(z)$ and $f_2(z)$ are not independent functions. The (broken) Ward identities associated with spatial translations can be expressed as[10]

$$\partial_\mu \langle T^{\mu j}(\tau, \vec{x}) \mathcal{P} \rangle_\beta = \delta^{(d-1)}(\vec{x}) D^j(\tau), \tag{34}$$

which can be used for $|\vec{x}| \neq 0$ to relate $f_1(z)$ and $f_2(z)$ via the differential equation (for $z > 0$)

$$-d\, f_1(z) + 4f_2(z) + z(f_1'(z) - 2f_2'(z)) = 0. \tag{35}$$

This leads to the following analytical solution for $f_2(z)$:

$$f_2(z) = c_1 z^2 + z^2 \int_0^z \mathrm{d}y\, \frac{-d\, f_1(y) + y f_1'(y)}{2y^3}, \tag{36}$$

where $c_1$ is an integration constant.

Combining the equation (35) with the OPE expansion (33) gives two equations for $f_1, f_2$ in terms of the data $\lambda_{T\widehat{\mathcal{O}}} \widehat{b}_{\widehat{\mathcal{O}}}$ and $\widehat{\Delta}$

$$
\begin{aligned}
f_1(z) &= \sum_{\widehat{\mathcal{O}}} \lambda_{T\widehat{\mathcal{O}}} \widehat{b}_{\widehat{\mathcal{O}}} \frac{2 - \widehat{\Delta}}{(d-2)(\widehat{\Delta} - 1)} z^{\widehat{\Delta}}, \\
f_2(z) &= \sum_{\widehat{\mathcal{O}}} \frac{\lambda_{T\widehat{\mathcal{O}}} \widehat{b}_{\widehat{\mathcal{O}}}}{2} \left(1 + \frac{d-1}{d-2} \frac{2 - \widehat{\Delta}}{\widehat{\Delta} - 1}\right) z^{\widehat{\Delta}}.
\end{aligned}
\tag{37}
$$

Asymptotic for $f_1$ and $f_2$ are given by

$$f_1(z) \overset{z \to 0}{\sim} \frac{2}{d-2} \lambda_{T\widehat{\mathbb{1}}}, \quad f_2(z) \overset{z \to 0}{\sim} -\frac{d/2}{d-2} \lambda_{T\widehat{\mathbb{1}}}, \quad f_1(z) \overset{z \to \infty}{\sim} \frac{b_T}{d} z^d, \quad f_2(z) \overset{z \to \infty}{\sim} 0, \tag{38}$$

where $\lambda_{T\widehat{\mathbb{1}}}$ is the defect one-point function of the stress-energy tensor (at zero temperature), while $b_T$ is the thermal one-point function of the defect (in absence of the defect). The first two equations in (38) correspond to the zero temperature limit, while the other two should be understood as the limit in which the defect is far away from the operators or equivalently the high temperature limit, thus reproducing the bulk thermal CFT.

The main motivation for studying the stress-energy tensor is its direct relation to the energy density of the system[11]

$$E(\vec{x}) = \frac{\mathcal{E}(z)}{|\vec{x}|^d} = -\langle T^{00}(\tau, \vec{x}) \mathcal{P} \rangle_\beta. \tag{39}$$

Since the system that we are considering is at finite temperature, it is possible to consider the thermodynamic equation

$$F(\vec{x}) = E(\vec{x}) - TS(\vec{x}) = E(\vec{x}) + T \frac{\mathrm{d}F(\vec{x})}{\mathrm{d}T}, \tag{40}$$

---

[10]See Appendix A for a derivation.

[11]The minus sign is conventional (see [16, 21]). It is there to ensure that the energy density is exactly equal to the stress-energy tensor one-point function after the appropriate Wick rotation.

which can be used to derive the free energy density $F(\vec{x})$ and the entropy density $S(\vec{x})$. We define the dimensionless quantity $\mathfrak{f}(z)$ through $F(\vec{x}) = |\vec{x}|^{-d}\mathfrak{f}(z)$, and (40) can be recast as

$$\mathfrak{f}(z) = \mathcal{E}(z) + z\mathfrak{f}'(z). \tag{41}$$

This is solved by

$$\mathfrak{f}(z) = c_2 z - z \int_1^z \frac{\mathrm{d}y}{y^2}\, \mathcal{E}(y), \tag{42}$$

where $c_2$ is an integration constant. The entropy density can be calculated by applying a derivative with respect to the temperature

$$S(\vec{x}) = -\frac{\mathrm{d}F}{\mathrm{d}T} = -\frac{1}{|\vec{x}|^{d-1}}\left(c_2 - \int_1^z \frac{\mathrm{d}y}{y^2}\mathcal{E}(y) - \frac{\mathcal{E}(z)}{z}\right). \tag{43}$$

In terms of CFT data,

$$F(\vec{x}) = \frac{1}{|\vec{x}|^d}\left(c_3 z + \sum_{\widehat{\mathcal{O}}} \lambda_{T\widehat{\mathcal{O}}}\, \widehat{b}_{\widehat{\mathcal{O}}}\, \frac{1}{\widehat{\Delta}-1} z^{\widehat{\Delta}}\right), \tag{44}$$

where $c_3$ is a function of CFT data to be determined below. The entropy density is

$$S(\vec{x}) = -\frac{1}{|\vec{x}|^{d-1}}\left(c_3 + \sum_{\widehat{\mathcal{O}}} \lambda_{T\widehat{\mathcal{O}}}\, \widehat{b}_{\widehat{\mathcal{O}}}\, \frac{\widehat{\Delta}}{\widehat{\Delta}-1} z^{\widehat{\Delta}-1}\right). \tag{45}$$

To fix $c_3$ we need to know the defect entropy at zero temperature. We notice that this limit requires that there are no defect operators (other than the identity) with $\widehat{\Delta} \leq 1$ and $\lambda_{T\widehat{\mathcal{O}}}\widehat{b}_{\widehat{\mathcal{O}}} \neq 0$, since otherwise the zero temperature limit is diverging. This condition can be explicitly verified: we perform this analysis in Appendix D. Finally, when the zero temperature limit of the entropy vanishes, then $c_3$ can be set to zero.

## 4.2 The free energy of a quark

When considering a Polyakov loop in gauge theory, its expectation value is known to be an order parameter for the confinement/deconfinement transition [49]. Although this transition is not the subject of this paper, it is important to note that the content of this paper can be easily adapted to the case of a Polyakov loop. In the following, we provide a definition and normalization of the loop that agrees with the literature, and show how to relate the free energy of an interacting quark to the OPE data.

Let us consider a non-Abelian gauge theory and define the operator

$$P = \frac{1}{N}\operatorname{Tr}\mathrm{P}\exp\left(\int_0^\beta \mathrm{d}\tau\, A_0(\tau)\right), \tag{46}$$

where P represents the path ordering. Correlation functions involving the operator above are naturally divergent, and therefore the expression (46) needs to be regularized. The most common regularization employed for conformal defects is to normalize any correlation function with the expectation value of the defect itself:

$$\langle \mathcal{O}_1(x_1)\cdots \mathcal{O}_n(x_n)\mathcal{P}\rangle = \frac{\langle \mathcal{O}_1(x_1)\cdots \mathcal{O}_n(x_n)P\rangle}{\langle P\rangle}. \tag{47}$$

In this convention, the finite-temperature expectation value $\langle \mathcal{P} \rangle_\beta$ is identically equal to the zero-temperature vacuum expectation value $\langle \langle \mathcal{P} \rangle \rangle = 1$.[12] This is in contrast with the requirement that the expectation value of Polyakov loops should be sensitive to the free energy $F_{\text{quark}}$ of the quark [11, 50, 51]:

$$\langle \mathcal{P} \rangle_\beta = e^{-\beta \left( F_{\text{quark}} - F_{\text{free quark}} \right)}, \tag{48}$$

where $F_{\text{free quark}}$ denotes the free energy of a free quark. To compensate for this discrepancy it is possible to define

$$\mathcal{P} = \frac{\operatorname{Tr} \operatorname{P} \exp \left( \int_0^\beta \mathrm{d}\tau \, A_0(\tau) \right)}{N \langle P_{\text{free}} \rangle_\beta}, \tag{49}$$

where the denominator is the expectation value of the Polyakov loop (46) at zero coupling. The definition above matches the expectation value (48) by construction and makes it a well-defined order parameter for the confinement/deconfinement transition: In fact, in the non-confining phase, $F_{\text{quark}} - F_{\text{free quark}}$ is expected to be finite and therefore $\langle \mathcal{P} \rangle_\beta$ is expected to be finite, while in the confining phase $F_{\text{quark}} - F_{\text{free quark}}$ is expected to be infinite, leading to $\langle \mathcal{P} \rangle_\beta = 0$. The limitation of this regularization is that a notion of $P_{\text{free}}$ is needed, i.e. a Lagrangian description is required. In this paper, we are interested in the most general setup and we will therefore use $\mathcal{P}$ for the regularized defect insertion. It should be noticed that in our work we are always considering the infinite-volume CFT at finite temperature where only the deconfining phase can be probed. To study the phase transition, we should either break conformality at zero temperature or place the theory on a spatial sphere at large $N$. In the latter case the Polyakov loop can still be considered a good order parameter provided that a more sophisticated analysis is conducted [52].

The expectation value of the Polyakov loop is given by the partition function of the full theory, divided by the partition function in the absence of the defect. In terms of free energies, this leads to

$$F_{\text{quark}} - F_{\text{free quark}} = \int \mathrm{d}^{d-1}x \left[ F(\vec{x}) - F_{\text{free}}(\vec{x}) - \frac{b_T}{d} \right], \tag{50}$$

where $b_T$ is the thermal one-point function of the stress-energy tensor and represents the finite temperature contribution to the free energy. Note that $F(\vec{x})$ can be related to defect one-point functions through Equation (44). Although we will not pursue this direction here, this analysis highlights the importance of further investigating temporal line defects from a bootstrap point of view as a tool to probe the confinement/deconfinement phase transition.

## 5 Applications

We present here applications of the concepts developed in the previous Sections for simple models, in which the computations can be carried out analytically. The first example is the case of generalized free fields, while the second one is the (interacting) $O(N)$ model in the $\varepsilon$-expansion and at large $N$.

### 5.1 Magnetic lines in generalized free field theory

To begin, we study the simple case of a magnetic line in Generalized Free Field (GFF) theory, in any spacetime dimension $d$. The thermal defect CFT is defined via the action

$$S = S_{\text{GFF}} + h \int_0^\beta \mathrm{d}\tau \, \phi(\tau, \vec{0}), \tag{51}$$

---

[12]Observe that divergences at finite temperature coincide with those at zero temperature, thus the expectation value of the defect defined in (46) is finite by construction.

where we denote as $\phi$ the fundamental field of the GFF theory and as $h$ the coupling constant of the defect. The defect in (51) is conformal if $\Delta_\phi = 1$, which corresponds to non-local theories in $2 \leq d < 4$ and a free scalar field theory in $d = 4$. Since the theory is free, it is possible to write the exact scalar propagator at finite temperature by using the method of images. We represent it as a Feynman diagrams using a solid black line

$$
\begin{aligned}
\circ\!\!-\!\!\!-\!\!\circ &= \langle \phi(\tau,\vec{x})\phi(0,\vec{0})\rangle_\beta \\
&= \frac{\pi}{2\beta|\vec{x}|}\left[\coth\left(\frac{\pi}{\beta}(|\vec{x}|-i\tau)\right) + \coth\left(\frac{\pi}{\beta}(|\vec{x}|+i\tau)\right)\right].
\end{aligned}
\tag{52}
$$

Finally, the defect itself is defined as

$$
\mathcal{P} = \frac{1}{\langle\mathcal{P}\rangle_\beta}\exp\left[-h\int_0^\beta d\tau\ \phi(\tau,\vec{0})\right].
\tag{53}
$$

In Feynman diagrams, we represent the defect with a red line. The corresponding Feynman rule is defined as

$$
\big)\!\!-\!\!\!-\ = -h\int_0^\beta d\tau.
\tag{54}
$$

### 5.1.1 One-point functions of bulk operators

We now compute the one-point functions of the bulk operators $\phi$ and $\phi^2$. This can be done straightforwardly by considering the Feynman diagrams of free theory.

**One-point function of $\phi$.** The one-point function of $\phi$ consists of a single diagram[13]

$$
\langle\phi(\tau,\vec{x})\mathcal{P}\rangle_\beta = \big)\!\!-\!\!\!-\!\!\circ
\tag{55}
$$

The empty circle on the right depicts the insertion of the operator $\phi(\tau,\vec{x})$ in the bulk. By inserting (52), it is easy to show that the diagram (55) exactly corresponds to the zero temperature one-point function of $\phi$, in the presence of the magnetic line

$$
\big)\!\!-\!\!\!-\!\!\circ = -h\int_0^\beta d\tilde{\tau}\ \langle\phi(\tau,\vec{x})\phi(\tilde{\tau})\rangle_\beta = -\frac{h\pi}{|\vec{x}|}.
\tag{56}
$$

By using the inversion formula (17) (or by simply comparing (56) to (9)), we find that only the defect identity $\widehat{\mathbb{1}}$ contributes to the defect OPE in this correlator. In fact, by using the equation of motion of $\phi$ in the free scalar theory in $4d$, it is possible to show that the only operators appearing in the defect expansion of $\phi$ are of the type $\widehat{\mathcal{O}}_{\widehat{\Delta}} = \partial^{i_1}\cdots\partial^{i_s}\widehat{\phi}(\tau)$ (i.e. $\widehat{\Delta} = s+1$). Since operators on the defect are, in the free case, restrictions of operators in the bulk, it is expected that $\langle\partial^{i_1}\cdots\partial^{i_s}\widehat{\phi}(\tau)\mathcal{P}\rangle_\beta = 0$, in agreement with the result in equation (56). In the case of a GFF the most general form of the operators appearing in the bulk to defect also includes for instance $\partial^{i_1}\cdots\partial^{i_s}\Box^k\phi$. Those operators share conformal dimensions with the operators discussed above, but they have different transverse spin: the conclusion in the GFF is that the sum over one-point functions of operators with the same conformal dimensions is zero.

---

[13]There is in principle an additional disconnected term, corresponding to the thermal one-point function of $\phi$. However, this term vanishes due to the $\mathbb{Z}_2$ symmetry of GFF in absence of the defect.

**One-point function of $\phi^2$.** It is also straightforward to calculate the one-point function of $\phi^2$. However, in this case there is a second diagram corresponding to the thermal one-point function $\langle \phi^2 \rangle_\beta$

$$\langle \phi^2(\tau, \vec{x}) \mathcal{P} \rangle_\beta = \qquad \qquad \qquad \qquad + \qquad \qquad \qquad \qquad \tag{57}$$

This expression can be made finite by considering the appropriate normal ordering. The value of the thermal one-point function can be found in Appendix C in [20] or in the discussion on GFF in [16]. This leads to

$$\langle \phi^2(\tau, \vec{x}) \mathcal{P} \rangle_\beta = \langle \phi^2(\tau, \vec{x}) \mathcal{P} \rangle + \langle \phi^2 \rangle_\beta = \frac{1}{|\vec{x}|^2} \left( h^2 \pi^2 + \frac{\pi^2}{6} z^2 \right), \tag{58}$$

where the first term can be calculated in the same way as (56). The expectation value independent from $\beta$ indicates the zero temperature correlator. We conclude that the defect operators contributing to the one-point function of $\phi^2$ are the defect identity $\widehat{\mathbb{1}}$ and a defect operator of dimension $\widehat{\Delta} = 2$. In GFF, since we choose $\Delta_\phi = 1$, the only two defect operators of dimension $\widehat{\Delta} = 2$ are $\widehat{\phi}^2$ and the displacement operator $D^i = \partial^i \phi$. $SO(d-1)$ symmetry prevents the latter from gaining a non-zero one-point function, and therefore we identify the second contribution in (58) as the one-point function of $\widehat{\phi}^2$ with defect one-point function

$$\langle \widehat{\phi}^2 \rangle_\beta = \langle \phi^2 \rangle_\beta = \frac{\pi^2}{6\beta^2}, \tag{59}$$

as expected.

### 5.1.2 The two-point function $\langle \phi\phi\mathcal{P} \rangle_\beta$

The two-point function of two elementary scalar fields $\phi$ in presence of the magnetic line reads

$$\langle \phi(\tau_1, \vec{x}_1)\phi(\tau_2, \vec{x}_2)\mathcal{P} \rangle_\beta = \qquad \qquad \qquad + \qquad \qquad \qquad$$
$$= \langle \phi(\tau_1, \vec{x}_1)\phi(\tau_2, \vec{x}_2) \rangle_\beta + \langle \phi(\tau_1, \vec{x}_1)\mathcal{P} \rangle_\beta \langle \phi(\tau_2, \vec{x}_2)\mathcal{P} \rangle_\beta. \tag{60}$$

It can be seen from equations (55) and (56) that the second term is independent of $\beta$, and thus all the thermal effects are contained in the disconnected term, corresponding to the two-point function without defect.

We now consider the collinear limit discussed in Section 3.1. In this case, the disconnected term can be deduced from 52 to be

$$\langle \phi(\tau, \vec{x})\phi(0, \vec{x}) \rangle_\beta = \frac{\pi^2}{\beta^2} \csc^2\left(\frac{\pi\tau}{\beta}\right) = \frac{2}{\beta^2} \sum_{k=0}^{\infty} (2k-1)\zeta_{2k}\left(\frac{\tau}{\beta}\right)^{2k-2}, \tag{61}$$

where $\zeta_{2k}$ is the Riemann $\zeta$-function evaluated at $2k$. This leads to the expression

$$\langle \phi(\tau, \vec{x})\phi(0, \vec{x})\mathcal{P} \rangle_\beta = \langle \phi(\tau, \vec{x})\phi(0, \vec{x})\mathcal{P} \rangle + \frac{1}{\beta^2} \sum_{k=1}^{\infty} c_k \left(\frac{\tau}{\beta}\right)^{2k-2}, \tag{62}$$

where the first term is the zero temperature expression, while $c_k = 2(2k-1)\zeta_{2k}$. This should be compared to the OPE expansion (5)

$$\langle \phi(\tau, \vec{x})\phi(0, \vec{x})\mathcal{P} \rangle_\beta = \langle \phi(\tau, \vec{x})\phi(0, \vec{x})\mathcal{P} \rangle + \sum_{\mathcal{O} \neq \mathbb{1}} \sum_{\widehat{\Delta} > 0} \mu_{\phi\phi\mathcal{O}} \lambda_{\mathcal{O}\widehat{\mathcal{O}}} \widehat{b}_{\widehat{\mathcal{O}}} \frac{\tau^{\Delta-2}}{\beta^{\widehat{\Delta}} |x|^{\Delta-\widehat{\Delta}}}. \tag{63}$$

The interpretation is that, from the operators appearing in this OPE, only the ones satisfying $\Delta = \widehat{\Delta}$ receive a finite temperature correction. More explicitly,

$$\mu_{\phi\phi\mathcal{O}}\lambda_{\mathcal{O}\widehat{\mathcal{O}}}\widehat{b}_{\widehat{\mathcal{O}}} = \begin{cases} c_{\Delta}, & \text{if } \Delta = \widehat{\Delta} \neq 0 \text{ and even,} \\ \mu_{\phi\phi\mathcal{O}}\lambda_{\mathcal{O}\widehat{\mathbb{1}}}, & \text{if } \widehat{\Delta} = 0, \\ 0, & \text{otherwise.} \end{cases} \tag{64}$$

It is straightforward to check analytically that these explicit results satisfy (23) for all odd $n$, providing a nice consistency check. This boils down to the fact that all odd derivatives of $\csc^2(x)$ at $x = \pi/2$ vanish.

We notice that in the two-point functions above, in the case of the free scalar theory in $4d$, the non-vanishing and $\beta$-dependent terms appear corresponding to the operators $\widehat{\phi}\,\partial^{i_1}\cdots\partial^{i_s}\widehat{\phi}$, which naturally appear in the bulk-to-defect OPEs of the bulk operators in the OPE $\widehat{\phi} \times \widehat{\phi}$. In the case of a GFF we have to add also the operators which vanish in the free theory because of equations of motions which are for instance $\widehat{\phi}\,\partial^{i_1}\cdots\partial^{i_s}\Box^k\widehat{\phi}$. As commented above, all those operators are degenerate in the conformal dimensions and differ for the value of the transverse spin. Therefore in the limit in which the two operators are at the same distance from the defect, since we are not sensitive to the transverse spin, the $\beta$-dependence is due to sum over operators of the same conformal dimensions.

### 5.1.3 Free energy and entropy

The one-point function of the stress-energy tensor can also be computed explicitly when the theory has a well-defined stress-energy tensor: this is again the case of the free scalar theory in $d = 4$.

The result for the free scalar theory is

$$\langle T^{\mu\nu}\mathcal{P}\rangle_{\beta} = \langle T^{\mu\nu}\mathcal{P}\rangle + \langle T^{\mu\nu}\rangle_{\beta}. \tag{65}$$

Specializing to the case $\mu = \nu = 0$, we have

$$\langle T^{00}\mathcal{P}\rangle_{\beta} = -\frac{1}{|\vec{x}|^4}\left(\frac{h^2\pi^2}{2} + \frac{2\pi^2}{15}z^4\right). \tag{66}$$

The value of the thermal one-point function of the stress-energy tensor for the free scalar theory in four dimensions is given, e.g., in [20] and [16]. As explained in Section 4.1, the one-point function of the stress-energy tensor is related to the energy density, the free energy density and the entropy density of the theory. By using (42) and (43), we find

$$F(\vec{x}) = \frac{1}{|\vec{x}|^4}\left(c_2 z + \frac{\pi^2}{90}(1-z)(45h^2 + 4z(1+z+z^2))\right). \tag{67}$$

Differentiating with respect to the temperature, we obtain the entropy density of the system

$$S(\vec{x}) = \frac{s(\vec{x})}{|\vec{x}|^3} = \frac{\pi^2\left(45h^2 + 16z^3 - 4\right) - 90c_2}{90|\vec{x}|^3}. \tag{68}$$

The limit $z \to 0$ produces

$$s(z) \overset{z\to 0}{\sim} \frac{\pi^2\left(45h^2 - 4\right) - 90c_2}{90}, \tag{69}$$

which is expected to be zero when we require the entropy to vanish at zero temperature.[14] This condition fixes $c_2$, and we end up with

$$F(\vec{x}) = \frac{\pi^2 \left(45h^2 - 4z^4\right)}{90|\vec{x}|^4}, \qquad S(\vec{x}) = \frac{8\pi^2 T^3}{45}. \tag{70}$$

Observe that, in the free theory, the entropy density does not depend on the defect. We do not expect this to carry on to the more general case of interacting theories, where terms involving the temperature and the defect coupling should mix. Moreover we observe that the free energy blows up at $|\vec{x}| \to 0$, as expected. Furthermore, it is a monotonic function of $z$ and its limit when $z \to \infty$ reproduces the free energy of the theory in the absence of the defect.

## 5.2 The magnetic line in the O($N$) model

We now turn our attention to the O($N$) model, to provide an example in which interactions are turned on, while computations can still be carried out analytically. At zero temperature, this theory was extensively studied without defect insertions [53–57], and the theory coupled to a localized magnetic line has been studied extensively [26, 31, 32, 58]. At finite temperature, the O($N$) model without defects is one of the most studied thermal CFTs [16–18, 59]. In this Section, we present two different but complementary approaches: the $\varepsilon$-expansion and the large $N$ analysis.[15]

### 5.2.1 $\varepsilon$-expansion

**The Wilson-Fisher fixed point.** The O($N$) model in $d$ dimensions, at finite temperature and in the presence of a magnetic line, is described by the action

$$S = \int_0^\beta \mathrm{d}\tau \int \mathrm{d}^{d-1}x \left[\frac{1}{2}(\partial_\mu \phi_i)^2 + \frac{\lambda}{4!}(\phi_i \phi_i)^2 + h\,\delta^{d-1}(\vec{x})\phi_1(\tau)\right]. \tag{71}$$

The action corresponds to a conformal defect for specific values of the coupling constants $\lambda$ and $h$. The idea of the $\varepsilon$-expansion is to set $d = 4 - \varepsilon$ and find the fixed point perturbatively as a function of $\varepsilon$. This fixed point is called the *Wilson-Fisher* fixed point, for which the coupling constants are known to take the following form [32, 58]

$$\frac{\lambda_\star}{(4\pi)^2} = \frac{3}{N+8}\varepsilon + \frac{9(3N+14)}{(N+8)^3}\varepsilon^2 + \mathcal{O}(\varepsilon^3), \tag{72}$$

$$h_\star^2 = N + 8 + \frac{4N^2 + 45N + 170}{2N + 16}\varepsilon + \mathcal{O}(\varepsilon^2), \tag{73}$$

such that the defect is conformal. Note that, at leading order, $\lambda_\star \sim \varepsilon$ and $h_\star \sim 1$. The Feynman rule associated to the quartic interaction in the bulk simply reads

$$\times = -\lambda \int_0^\beta d\tau \int d^{d-1}x, \tag{74}$$

while the thermal bulk propagator in arbitrary dimensions is

$$\circ\!\!-\!\!\!-\!\!\!-\!\!\circ = \frac{\Gamma(d/2 - 1)}{4\pi^{d/2}} \sum_{m\in\mathbb{Z}} \frac{1}{(|\vec{x}|^2 + |\tau + m\beta|^2)^{d/2-1}}. \tag{75}$$

---

[14]The standard zero-temperature definition of entropy is proportional to the logarithm of the number of ground states and therefore zero in free theory by uniqueness of the ground state.

[15]We are grateful to Simone Giombi for interesting discussions regarding this Section.

**One-point function of $\phi_1$.** At order $\mathcal{O}(\varepsilon)$, the one-point function of $\phi_1$ is given by the following Feynman diagrams

$$\langle\,\phi_1(\tau,\vec{x})\mathcal{P}\,\rangle_\beta = \text{(diagram)} + \text{(diagram)} + \text{(diagram)} + \mathcal{O}(\varepsilon^2). \tag{76}$$

To compute these diagrams, we borrow the conventions for the normalization of the vertices and of the propagators from [32] and we make use of the master integrals

$$\int_{-\infty}^{\infty} \frac{d\tau}{(|\vec{x}|^2 + \tau^2)^{1-\varepsilon/2}} = \frac{\sqrt{\pi}\,\Gamma\left(\frac{1-\varepsilon}{2}\right)}{\Gamma(1-\varepsilon/2)|\vec{x}|^{1-\varepsilon}}, \tag{77}$$

$$\int \frac{d^{d-1}x_2}{|\vec{x}_{12}|^a|\vec{x}_2|^b} = \frac{\Gamma\left(\frac{d-1-a}{2}\right)\Gamma\left(\frac{d-1-b}{2}\right)\Gamma\left(\frac{a+b+1-d}{2}\right)}{\Gamma\left(\frac{a}{2}\right)\Gamma\left(\frac{b}{2}\right)\Gamma\left(\frac{2(d-1)-a-b}{2}\right)} \frac{\pi^{(d-1)/2}}{|\vec{x}_1|^{a+b+1-d}}. \tag{78}$$

The first diagram is similar to (56)

$$\text{(diagram)} = -\frac{1}{4}h\pi^{\frac{\varepsilon-3}{2}}\Gamma\left(\frac{1-\varepsilon}{2}\right)\frac{1}{|\vec{x}|^{1-\varepsilon}}. \tag{79}$$

The second diagram in (76) does not appear at zero temperature (as massless tadpoles vanish), and can be viewed as a manifestation of the *thermal mass*. At order $\mathcal{O}(\varepsilon)$, the thermal mass is given by [60–63]

$$m_{\text{th}}^2 = \frac{\varepsilon N}{2\beta^2(N+8)} + \mathcal{O}(\varepsilon^2), \tag{80}$$

and the diagram is equal to

$$\text{(diagram)} = -\frac{1}{16}hm_{\text{th}}^2\pi^{\frac{\varepsilon-3}{2}}\Gamma\left(-\frac{1+\varepsilon}{2}\right)|\vec{x}|^{1+\varepsilon}. \tag{81}$$

The third diagram also appears at zero temperature, and does not receive any thermal correction

$$\text{(diagram)} = -\frac{1}{768}h^3\lambda\ \pi^{\frac{3(\varepsilon-3)}{2}}\frac{\Gamma^3\left(\frac{1-\varepsilon}{2}\right)}{\varepsilon(1-3\varepsilon)}\frac{1}{|\vec{x}|^{1-3\varepsilon}}. \tag{82}$$

This expression (82) is divergent in the $\varepsilon \to 0$ limit and requires renormalization. We follow the procedure as explained in [32, 58].

Putting everything together, we obtain

$$\frac{\langle\,\phi_1(\vec{x})\mathcal{P}\,\rangle_\beta}{\sqrt{\langle\,\phi_1(\infty)\phi_1(0)\,\rangle}} = -\frac{\sqrt{N+8}}{2|\vec{x}|}\left[1 + \left(\frac{N^2 - 3N - 22 + 2(N+8)^2\log(2|\vec{x}|)}{4(N+8)^2} + \frac{N}{4(N+8)}z^2\right)\varepsilon \right.$$
$$\left. + \mathcal{O}(\varepsilon^2)\right], \tag{83}$$

where $\langle\,\phi_1(\infty)\phi_1(0)\,\rangle$ on the left-hand side refers to the normalization of the two-point function, and $z$ is as usual the temperature-dependent cross-ratio defined in (8).

**OPE data.** The first and second terms in (83) correspond to the zero temperature result, with $\log|\vec{x}|$ coming from the expansion of $\Delta_\phi$ at order $\varepsilon$ and the it matches the results of [32, 58]. The $z$-dependent (hence, $\beta$-dependent) terms are induced by the thermal mass. The powers of $z$ indicate the presence, at this order, of one defect operator of bare $1d$ scaling dimension $\widehat{\Delta} = 2$. Since the only two operators with the latter bare dimension are $\widehat{t}^{\,2}$, the square of

the tilt operator $\widehat{t}_a = \widehat{\phi}_{a \neq 1}$, and the displacement operator $D_i$, we have that only the first one contributes since the displacement operator one-point function vanishes because of $SO(d-1)$ symmetry. Following the notation of (12), we extract the OPE data

$$\lambda_{\phi_1 \widehat{t}^2} \, \widehat{b}_{\widehat{t}^2} = -\frac{N}{8\sqrt{N+8}} \varepsilon + \mathcal{O}\left(\varepsilon^2\right). \tag{84}$$

This is a new prediction which we will also use in the subsequent Section, after having performed the large $N$ analysis.

### 5.2.2 Large $N$ analysis

We now consider the same system, but in the limit $N \to \infty$. In this regime, it is possible to normalize the coupling constants, such that the action (71) can be expressed as

$$S = \int_0^\beta d\tau \int d^{d-1}x \left[ \frac{1}{2}(\partial_\mu \phi_i)^2 + \frac{\widetilde{\lambda}}{N}(\phi_i \phi_i)^2 + \sqrt{N}\widetilde{h}\,\delta^{d-1}(\vec{x})\phi_1(\tau) \right]. \tag{85}$$

Note that, in these conventions, we have $\widetilde{\lambda}_\star \sim \mathcal{O}(1)$ and $\widetilde{h}_\star \sim \mathcal{O}(1)$ [58]. The Hubbard-Stratonovich transformation of the action gives

$$S = \int_0^\beta d\tau \int d^{d-1}x \left[ \frac{1}{2}(\partial_\mu \phi_i)^2 + \frac{1}{2}\sigma \phi_i \phi_i + \sqrt{N}\widetilde{h}\,\delta^{d-1}(\vec{x})\phi_1(\tau) \right]. \tag{86}$$

Integrating out the fields $\phi_i$, we obtain the effective action for $\sigma$

$$S_{\text{eff}}[\sigma] = \frac{N}{2}\text{Tr}\log(-\Box + \sigma) - \frac{\widetilde{h}^2 N}{2} \int_{\mathcal{M}_\beta} d^d x \int_{\mathcal{M}_\beta} d^d y\, \delta^{d-1}(\vec{x})\left(\frac{1}{-\Box + \sigma}\right)(x-y)\delta^{d-1}(\vec{y}). \tag{87}$$

In principle, the extremization of (87) yields the one-point function $\langle \sigma(\vec{x})\mathcal{P} \rangle_\beta$. In absence of the defect ($\widetilde{h} = 0$), this procedure is easy to apply, and results in the thermal mass at large $N$ [16, 64]. In the presence of the defect, this computation is more involved, even at zero temperature [2], but a combination of the equations of motion and the constraints from conformal symmetry can be used to derive $\langle \sigma(\vec{x})\mathcal{P} \rangle_\beta$ [58].

A similar approach can be followed in the finite temperature case. From the action (87), we obtain, as a consequence of the equations of motion, the following differential equation:

$$\left( \frac{\partial^2}{\partial |\vec{x}|^2} + \frac{d-2}{|\vec{x}|}\frac{\partial}{\partial |\vec{x}|} - \langle \sigma(\vec{x})\mathcal{P} \rangle_\beta \right)\langle \phi_1(\vec{x})\mathcal{P} \rangle_\beta = 0, \tag{88}$$

for $|\vec{x}| \neq 0$. In the large $N$ limit, the scaling dimensions of $\phi_1$ and $\sigma$ are known to be [58]

$$\Delta_{\phi_1} = \frac{d-2}{2} + \mathcal{O}\left(\frac{1}{N}\right), \qquad \Delta_\sigma = 2 + \mathcal{O}\left(\frac{1}{N}\right). \tag{89}$$

The defect spectrum in the bulk-defect OPE of the operators $\phi_2, \ldots, \phi_N$ is known to all orders in $\varepsilon$ at large $N$, while the one for $\phi_1$ is not known to the best of our knowledge. Nonetheless, it is possible to use the known defect spectrum [58] to write the first few terms in the OPE, corresponding to a low-temperature expansion of the correlator. The thermal one-point function of $\phi_1$ in the presence of the defect is then

$$\langle \phi_1(\vec{x})\mathcal{P} \rangle_\beta = \frac{1}{|\vec{x}|^{\frac{d}{2}-1}}\left[ c_0 + c_1 z + c_2 z^{1+\gamma} + c_3 z^2 + \mathcal{O}\left(z^{2+\epsilon}\right) \right], \tag{90}$$

at least in $3 \leq d \leq 4$, where the first operator corresponds to the identity $\widehat{\mathbb{1}}$, the second is the contribution of the tilt operator $\widehat{t}_a$, the third term is due to $\widehat{\phi}_1$ and the fourth to the composite operator constructed out of the tilt operator $\widehat{t}^2$. $\widehat{\Delta} = 2 + \epsilon$, with $\epsilon > 0$, corresponds to the conformal dimension of the next lightest operator in the spectrum of the defect CFT. In the OPE (90), $\gamma$ is the anomalous dimensions of $\widehat{\phi}_1$, which in $d = 3$ is $\gamma_{3d} = 0.541728\ldots$ [58]; the coefficients $c_i$ are the OPE coefficients defined as in the equation (12). It should be noted that there are other operators in the spectrum, for instance, $\partial^i \widehat{\phi}_1$: however, one-point functions of operators with odd transverse spin vanish due to SO$(d-1)$ symmetry, so they do not contribute. Furthermore $c_1 = 0$ since the tilt operator is an O$(N-1)$ vector and its one-point function vanishes for symmetry reasons. It is still instructive to keep this term which will not appear in any case in the following. The value of the identity contribution is fixed by the zero temperature behaviour of the correlator to be $c_0 = \lambda_{\phi_1 \hat{1}}$: in $d = 3$, its numerical value is $(c_{3d,0})^2 = N \times 0.558113\ldots$ [58].

Plugging the OPE (90) in the equation of motion (88), we obtain an expansion for the one-point function of the Hubbard-Stratonovich field:

$$\langle \sigma(\vec{x}) \mathcal{P} \rangle_\beta = \frac{1}{|\vec{x}|^2} \left[ \frac{(d-2)(4-d)}{4} + \frac{c_2}{c_0} \gamma(1+\gamma) z^{1+\gamma} + 2\frac{c_3}{c_0} z^2 + o\left(z^2\right) \right]. \qquad (91)$$

The first term in (91) corresponds to the zero temperature result [58], while the second term is the first non-trivial thermal correction, triggered by the defect operator $\widehat{\phi}_1$. We see that the operator of dimension $\widehat{\Delta} = 2$ contributes to the one-point functions of $\sigma$, too. The latter corresponds also to the tree level contribution to $\langle \phi_i \phi_i \mathcal{P} \rangle_\beta$, i.e. order 0 in epsilon.

To fully fix the two one-point functions (90) and (91) at the order $\mathcal{O}(z^2)$ we only need to fix $c_2$ and $c_3$. These coefficients are generically hard to compute, even at large $N$. In principle they can be obtained from minimizing the effective action (87). We will not pursue this direction here, and instead we give an estimation of the coefficients using the results obtained in the previous Section using the $\varepsilon$-expansion. By comparing (90) with (83), we find

$$c_2 = \lambda_{\phi_1 \widehat{\phi}_1} \widehat{b}_{\widehat{\phi}_1} = \mathcal{O}(\varepsilon^2), \qquad c_3 = \lambda_{\phi_1 \widehat{t}^2} \widehat{b}_{\widehat{t}^2} = -\frac{\sqrt{N}}{2} \varepsilon + \mathcal{O}\left(\varepsilon^2\right). \qquad (92)$$

Let us also comment that the equations of motion relate OPE coefficients in $\langle \sigma \mathcal{P} \rangle_\beta$ with OPE coefficients in $\langle \phi_1 \mathcal{P} \rangle_\beta$. Comparing the two expressions we are also able to extract relations between zero temperature bulk to defect coefficients, in particular $\lambda_{\sigma \widehat{\phi}_1} = \gamma(1+\gamma) \lambda_{\phi_1 \widehat{\phi}_1} / \lambda_{\phi_1 \hat{1}}$ (this equation only holds if $\widehat{b}_{\widehat{\phi}_1} \neq 0$) and $\lambda_{\sigma \widehat{t}^2} = 2\lambda_{\phi_1 \widehat{t}^2} / \lambda_{\phi_1 \hat{1}}$. Those relations are not simple to see at zero temperature since the one-point functions of local defect operators are zero (apart from the identity).

## 6 Conclusions and outlook

In this paper, we initiated the study of defect CFT at finite temperature from a bootstrap perspective. We specialized our analysis to the case of a temporal conformal line defect wrapping the thermal circle, for which one important example is the Polyakov loop in gauge theories. We use the symmetries of the theory to identify the OPE data necessary to determine the correlation functions of local operators in the presence of the defect, which consist of (1) the zero temperature data, (2) the thermal one-point functions of bulk operators, and (3) the new defect one-point functions. As we emphasised in Section 2.2.2, one essential aspect is that the defect one-point functions should not be viewed as the one-point functions of a thermal CFT; rather, they are induced by thermal effects in the bulk.

Local properties of the theory at zero temperature are preserved at finite temperature for a finite radius of convergence. This allows us to use the bulk and defect OPEs to derive an inversion formula for the defect one-point functions in terms of the bulk one-point function. The KMS condition imposes non-trivial consistency constraints on the two-point functions of identical bulk operators, which we use to obtain sum rules involving the new defect one-point functions coefficients. These relations are very similar to the sum rules that involve the thermal one-point functions in the absence of a defect [16]. Moreover, we make use of the channel duality given by the KMS condition to estimate the asymptotic behaviour of those coefficients. We also relate the free energy and the entropy density of the system to the one-point function of the stress-energy tensor and discuss the free energy of a moving quark.

The formalism described above is tested in two concrete examples, where calculations can be performed explicitly. The first one is the case of generalized free scalar field theory, for which we computed analytically the one-point functions of $\phi$ and $\phi^2$ through Wick contractions. The diagrams can be interpreted as zero temperature results, to which we add (at most) products of shorter correlators involving the thermal one-point functions. We also provide the two-point function of elementary scalars as a check of the sum rules. The second example is the $O(N)$ model, which is an interacting CFT, both in the context of the $\varepsilon$-expansion and in the large $N$ limit. In the first case, we calculate the one-point function of $\phi_1$ and notice that, at first order in $\varepsilon$, only one diagram (corresponding to the contribution of the thermal mass) is temperature-dependent. This allows us to extract the relevant OPE data. At large $N$, the system can be studied using the Hubbard-Stratonovich formulation of the $O(N)$ model. In this case, we find the form of the one-point function of $\sigma$, up to a few coefficients that can be estimated numerically for $d \sim 4$ using the results of the $\varepsilon$-expansion. These results can, in principle, be tested and complemented using diverse methods, such as lattice and Monte-Carlo simulations. It would also be interesting to compare the kinematical structure of the OPE with a suitable generalization of the work of [65, 66].

There are many interesting directions that can be further explored:

- ⋆ The similarities of the bootstrap problem discussed in this paper with the one of [16] suggest that analogous techniques can be used to solve for the unknown coefficients. These were used to obtain thermal one-point functions of the 3$d$ Ising model [19], and the results are in good agreement with the lattice estimations. We believe that this approach can be extended to the KMS consistency conditions arising in the defect setup, and we are currently exploring this direction for the case of $\mathcal{N} = 4$ SYM with a supersymmetric Wilson line, where the bootstrap program provides a rich set of zero temperature data both numerically [22, 29, 67–71] and analytically [27, 72–77]. We hope to report soon on new results [78].

- ⋆ In holographic theories, studied in the bulk recently in, e.g., [79–84], a phase transition is expected to occur at large $N$ in the geometry $S^1_\beta \times S^{d-1}_R$ for a certain critical value of the ratio $\beta/R$. In the gravitational dual, this corresponds to a phase transition between thermal excitations of the bulk theory in the presence of a black hole, and for which the Polyakov loop is the order parameter [12, 85]. It is important for this reason to extend the tools developed in this paper to the *finite volume* case. In particular, the free energy should play an important role, since it can also be used as a probe for the phase transition.

- ⋆ In this work, we considered line defects wrapping the thermal circle. A natural extension would be to consider line defects extending in one of the spatial directions (see Fig. 1). Even though these setups do not have a direct connection to confinement, they are still very interesting. This study could also be generalized to defects with a lower codimension, such as surfaces and boundaries.

⋆ In this paper, we limited our analysis to correlation functions of local operators. An interesting extension is to consider correlation functions of defects themselves. The most accessible setup might be the case in which two parallel defects of the type studied in this paper are placed at a distance $L$ from one another. When the system is probed at a large distance (with respect to the two defects), we expect the defects to *fuse* in an OPE for extended operators. This is a largely unexplored area of research, although recent progress has taken place at zero temperature [86–90]. For the case of two Polyakov loops, the expectation is that the correlator of two defects reproduces the exponential behavior between two quarks

$$\langle \mathcal{P}(0)\mathcal{P}(L)\rangle_\beta = e^{-\beta V(L)}\big[\, 1 + \mathcal{O}\big(e^{-\beta \mathcal{E}}\big)\big], \tag{93}$$

with $\mathcal{E}$ the energy difference between the potential and the first excited state. Setting up an OPE both at short and long (as for instance in [91]) distances for this type of observables would be of great interest, and would allow to extend our analysis to this type of system.

⋆ Recently, many interesting features of thermal CFTs were studied by using an effective field theory (EFT) approach [90, 92–95]. It might be possible to extend this approach to setups which involve a defect operator. Furthermore, bootstrap and EFT approaches have also been recently proposed for the study of moduli spaces of CFTs [38,96]. The $1d$ theory living on the defect in our setup is in a similar phase, and it could be an interesting target for future work in this direction. In the case of (generalized) free theories, it would be interesting to see whether a connection with zero temperature conformal graphs (as in [97, 98]) persists for the one-point functions of defect operators introduced in this paper.

# Acknowledgments

It is a pleasure to thank António Antunes, Simone Giombi, Apratim Kaviraj, Diego Rodríguez-Gómez and Volker Schomerus for very useful discussions.

**Funding information** JB and EP are supported by ERC-2021-CoG - BrokenSymmetries 101044226. AM, JB and EP have benefited from the German Research Foundation DFG under Germany's Excellence Strategy – EXC 2121 Quantum Universe – 390833306. BF's research supported by the State Agency for Research of the Spanish Ministry of Science and Innovation through the "Unit of Excellence María de Maeztu 2020-2023" award to the Institute of Cosmos Sciences (CEX2019-000918-M), by grants PID2019-105614GB-C22 and PID2022-136224NB-C22, funded by MCIN/AEI/ 10.13039/501100011033/FEDER, UE and by AGAUR, grant 2021 SGR 00872. The authors would like to express special thanks to the Mainz Institute for Theoretical Physics (MITP) of the Cluster of Excellence PRISMA⁺ (project ID 39083149), where our collaboration began, for its hospitality and support. AM and JB thank the Hamilton Mathematics Institute, Trinity College Dublin and the organizers of the workshop *CFT and Holography: Heavy and Thermal States and Black Holes*, where this work was finalized and presented for the first time.

# A   Broken Ward identities

We derive (broken) Ward identities for broken and un-broken generators following the procedure of [20], where many details are given. The main idea is that, since the conformal

symmetry is expected to be non-anomalous, the action of the theory has to be non-invariant under the broken generators. Making use of the path integral formulation, it is possible to derive how a symmetry generator acts on a correlation function and relate it to the variation of the action.

More concretely, we can consider the following formal action as a starting point

$$S = S_{\text{bulk}} + h \int_0^\beta d\tau \, \phi(\tau, \vec{0}), \tag{A.1}$$

where we included a term describing a line wrapping the thermal circle located at the spatial origin.[16] The action $S_{\text{bulk}}$ is simply the action in absence of the defect. We have that, for any non-anomalous, infinitesimal symmetry transformation $\delta\mathcal{O} = i\omega^a \boldsymbol{G}_a \mathcal{O}$, the following formal Ward identity holds

$$\delta\langle \mathcal{O}_1(x_1)\cdots\mathcal{O}_n(x_n)\mathcal{P}\rangle_\beta = \langle\delta S_{\text{bulk}}\, \mathcal{O}_1(x_1)\cdots\mathcal{O}_n(x_n)\mathcal{P}\rangle_\beta. \tag{A.2}$$

Isolating the variation of the Polyakov loop we obtain

$$\langle\delta\left(\mathcal{O}_1(x_1)\cdots\mathcal{O}_n(x_n)\right)\mathcal{P}\rangle_\beta + \langle\mathcal{O}_1(x_1)\cdots\mathcal{O}_n(x_n)\delta\mathcal{P}\rangle_\beta = \langle\delta S_{\text{bulk}}\, \mathcal{O}_1(x_1)\cdots\mathcal{O}_n(x_n)\mathcal{P}\rangle_\beta. \tag{A.3}$$

We can expand the defect using the definition (A.1)

$$\langle\delta\left(\mathcal{O}_1(x_1)\cdots\mathcal{O}_n(x_n)\right)\mathcal{P}\rangle_\beta - h\,\delta\int_0^\beta d\tau\langle\mathcal{O}_1(x_1)\cdots\mathcal{O}_n(x_n)\phi(\tau,\vec{0})\mathcal{P}\rangle_\beta$$
$$= \langle\delta S_{\text{bulk}}\, \mathcal{O}_1(x_1)\cdots\mathcal{O}_n(x_n)\mathcal{P}\rangle_\beta. \tag{A.4}$$

Following the procedure in [20], we obtain the following result

$$i\sum_i\langle\mathcal{O}_1(x_1)\cdots\boldsymbol{G}_a\mathcal{O}_i(x_i)\cdots\mathcal{O}_n(x_n)\mathcal{P}\rangle_\beta = \int d^{d-1}y\,\langle\Gamma_a^{\mathcal{P},\beta}(\vec{y})\mathcal{O}_1(x_1)\cdots\mathcal{O}_n(x_n)\mathcal{P}\rangle_\beta, \tag{A.5}$$

where we introduced a new breaking term $\Gamma_a^{\mathcal{P},\beta}$, defined as a defect-induced correction to the thermal breaking term $\Gamma_a^\beta$

$$\Gamma_a^{\mathcal{P},\beta}(\vec{y}) = \Gamma_a^\beta(\vec{y}) + h\,\delta(\vec{y})\,\boldsymbol{G}_a\int_0^\beta d\tau\,\phi(\tau,\vec{y}). \tag{A.6}$$

**Broken translations.**    Let us focus on the broken Ward identity associated with translations. We know that the purely thermal breaking term is trivially $\Gamma_\mu^\beta = 0$, hence the breaking is triggered only by the defect

$$\Gamma_\mu^{\mathcal{P},\beta}(\vec{y}) = h\,\delta(\vec{y})\int_0^\beta d\tau\,(\partial_\mu\phi)(\tau,\vec{y}). \tag{A.7}$$

If we assume the field $\phi$ to be periodic over the thermal circle, the breaking term simplifies

$$\Gamma_0^{\mathcal{P},\beta} = 0, \qquad \Gamma_i^{\mathcal{P},\beta}(\vec{y}) = h\,\delta(\vec{y})\int_0^\beta d\tau\,D_i(\tau,\vec{y}), \tag{A.8}$$

---

[16]The field $\phi$ is not necessarily a fundamental field of the theory, but an operator of conformal dimension one.

where $D_i$ is the displacement operator. We conclude that, as expected, translations along the thermal circle are preserved, but spatial translations are broken and their Ward identity is

$$\sum_i \langle \mathcal{O}_1(x_1) \cdots \partial_i \mathcal{O}_i(x_i) \cdots \mathcal{O}_n(x_n) \mathcal{P} \rangle_\beta = h \int_0^\beta d\tau \left\langle D_i(\tau, \vec{0}) \mathcal{O}_1(x_1) \cdots \mathcal{O}_n(x_n) \mathcal{P} \right\rangle_\beta . \tag{A.9}$$

The unintegrated broken Ward identity can be rewritten operatorially as

$$\partial_\mu T^\mu{}_i(\tau, \vec{y}) = \delta(\vec{y}) D_i(\tau, \vec{y}), \tag{A.10}$$

which is well known from the first studies of defects in CFT [7]. Finite temperature effects do not bring any contribution to the breaking term.

**Broken dilatations.** We now study broken dilatations as an example of a symmetry which is broken both by thermal effects and by the introduction of a defect. The purely thermal breaking term is $\Gamma^\beta = \beta T^{00}$, so

$$\Gamma^{\mathcal{P},\beta}(\vec{y}) = \beta T^{00}(0, \vec{y}) + h \, \delta(\vec{y}) D \int_0^\beta d\tau \, \phi(\tau, \vec{y}). \tag{A.11}$$

The action of the dilatation operator is

$$\begin{aligned}
D \int_0^\beta d\tau \, \phi(\tau, \vec{y}) &= \int_0^\beta d\tau \, \left( \tau \partial_\tau + y^i \partial_i + \Delta_\phi \right) \phi(\tau, \vec{y}) \\
&= \left( \Delta_\phi - 1 \right) \int_0^\beta d\tau \, \phi(\tau, \vec{y}) + y^i \int_0^\beta d\tau \, D_i(\tau, \vec{y}) + \beta \phi(0, \vec{y}),
\end{aligned} \tag{A.12}$$

and the broken Ward identity reads

$$\begin{aligned}
D \langle \mathcal{O}_1(x_1) \cdots \mathcal{O}_n(x_n) \mathcal{P} \rangle_\beta &= \beta \int d^{d-1} y \left\langle \left( T^{00}(0, \vec{y}) + h \, \delta(\vec{y}) \phi(0, \vec{y}) \right) \mathcal{O}_1(x_1) \cdots \mathcal{O}_n(x_n) \mathcal{P} \right\rangle_\beta \\
&\quad + h \left( \Delta_\phi - 1 \right) \int_0^\beta d\tau \, \langle \phi(\tau, \vec{y}) \mathcal{O}_1(x_1) \cdots \mathcal{O}_n(x_n) \mathcal{P} \rangle_\beta .
\end{aligned} \tag{A.13}$$

Interestingly, when the defect is conformal $\Delta_\phi = 1$ we witness a simplification

$$D \langle \mathcal{O}_1(x_1) \cdots \mathcal{O}_n(x_n) \mathcal{P} \rangle_\beta = \beta \int d^{d-1} y \left\langle \left( T^{00}(0, \vec{y}) + h \, \delta(\vec{y}) \phi(0, \vec{y}) \right) \mathcal{O}_1(x_1) \cdots \mathcal{O}_n(x_n) \mathcal{P} \right\rangle_\beta . \tag{A.14}$$

This broken Ward identity can be rewritten operatorially as

$$D = -\beta \left( H + E_0 + h \boldsymbol{\phi} \right), \tag{A.15}$$

where we introduced the thermal Hamiltonian and the thermal ground state [20]

$$H = -\int d^{d-1} y \left( T^{00}(\tau, \vec{y}) + E_0 \right), \quad E_0 = \frac{d-1}{d} \frac{b_T}{\beta^d}. \tag{A.16}$$

In the case of the dilatations, defect and thermal effect combine to produce a correction to the Hamiltonian. It should be noted that the equation (A.15) is strictly valid for conformal defects that admit a Lagrangian realization as in (A.1).

The definition (A.1) is not the most general definition of conformal line defect, for instance the case of disorder-type defect is usually defined via boundary conditions. Nonetheless the equations which are only based on the symmetries (e.g. (A.9)) at zero temperature are expected to remain valid.

# B   A comment on the real time formalism

In this Appendix, we explain why, although the zero temperature data is in principle sufficient to compute all the thermal correlation functions, a bootstrap approach is still favourable.

Let us start by considering one-point functions on the geometry $\mathbb{R} \times S_R^{d-1}$ and in absence of the defect. Radial quantization fixes the Hamiltonian of the theory to be

$$H = \frac{D}{R}, \tag{B.1}$$

where $D$ is the generator of dilatations. Since thermal effects can be considered by introducing the density matrix $\rho = e^{-\beta H}$, one-point functions and the partition function can be written as

$$\langle \mathcal{O} \rangle_{\beta,R} = \frac{1}{\mathcal{Z}} \sum_{\mathcal{O}'} e^{-\frac{\beta}{R}\Delta_{\mathcal{O}'}} \langle \mathcal{O}' | \mathcal{O} | \mathcal{O}' \rangle, \quad \mathcal{Z} = \sum_{\mathcal{O}'} e^{-\frac{\beta}{R}\Delta_{\mathcal{O}'}}. \tag{B.2}$$

Schematically, the above equation implies that [15]

$$\langle \mathcal{O} \rangle_\beta \sim \lim_{R \to \infty} \frac{1}{\beta^{\Delta_{\mathcal{O}}}} \frac{\sum_{\mathcal{O}'} C_{\mathcal{O}'\mathcal{O}'\mathcal{O}} e^{-\frac{\beta}{R}\Delta_{\mathcal{O}'}}}{\sum_{\mathcal{O}'} e^{-\frac{\beta}{R}\Delta_{\mathcal{O}'}}}, \tag{B.3}$$

where the limit decompactifies the spatial sphere and reproduces the thermal one-point function on the thermal geometry $S_\beta^1 \times \mathbb{R}^{d-1}$. The equation above can even be expressed in terms of conformal blocks by summing only over primaries [99, 100]. From that expression, one can try to take the limit, and in two dimensions some results are available in Appendix B of [16].

A similar procedure can be followed when a defect is introduced. In fact,

$$\langle \mathcal{O}\mathcal{P} \rangle_{\beta,R} = \frac{1}{\mathcal{Z}_\mathcal{P}} \sum_{\mathcal{O}'} e^{-\frac{\beta}{R}\Delta_{\mathcal{O}'}} \langle \mathcal{O}' | \mathcal{O}\mathcal{P} | \mathcal{O}' \rangle, \quad \mathcal{Z}_\mathcal{P} = \sum_{\mathcal{O}'} e^{-\frac{\beta}{R}\Delta_{\mathcal{O}'}} \langle \mathcal{O}' | \mathcal{P} | \mathcal{O}' \rangle. \tag{B.4}$$

In principle, one can trade the thermal one-point function in the presence of the defect on the thermal manifold with an infinite sum over zero temperature three-point functions in the presence of the defect. Since conformal symmetry is broken, the three-point functions are generally difficult to compute and even their kinematical structure can be intricate. For this reason, in particular, in the presence of the defect, the bootstrap approach established in this paper is expected to be more efficient in practical situations: one-point functions of the defect operators are now seen as *new* data and they are the object of a bootstrap problem. Nonetheless, if one is interested in the geometry $S_\beta^1 \times S_R^{d-1}$, the real-time formalism provides an efficient expansion in the variable $e^{-\beta/R}$ in which light operators dominate. In this situation, it would be interesting to compare the approach of this paper with the real time formalism and maybe gain new insights from combining them. We leave this interesting direction for future study.

# C   Defect thermodynamics in $d = 2$

We discuss here the defect thermodynamics in $d = 2$ since there are important differences between this case and $d > 2$. In particular the Ansatz (31) and (32) simplifies drastically since there is only one possible symmetric and invariant tensor structure in $d = 2$

$$\langle T^{11}\mathcal{P} \rangle_\beta = -\langle T^{00}\mathcal{P} \rangle_\beta = \frac{f(z)}{|x_1|^2}, \qquad \langle T^{10} \rangle_\beta = \langle T^{01} \rangle_\beta = 0. \tag{C.1}$$

Furthermore the conservation of the stress-energy tensor, $\partial_1 \langle T^{11} \mathcal{P} \rangle_\beta = 0$, implies

$$z f'(z) - 2 f(z) = 0, \tag{C.2}$$

which is solved by $f(z) = c_1 z^2$, with $c_1$ being an integration constant. Thus, in $d = 2$, the energy density is position independent. On the other hand, comparing with the discussion in Section 2.2, we interpret this result as the fact that only defect operators of conformal dimensions $\widehat{\Delta} = 2$ contribute to the one-point function of the stress-energy tensor. It is possible to integrate the energy density and use basic thermodynamic relations to compute the entropy of the system

$$S = -2 \frac{c_1}{\beta} \text{Vol}(\mathbb{R}) + \text{const.} \tag{C.3}$$

This is consistent with the expected result in two dimensions: in fact, modular invariance fixes the entropy to be [13, 101, 102]

$$S = \frac{\pi c}{3\beta} \text{Vol}(\mathbb{R}) + \text{const.}, \tag{C.4}$$

where $c$ is the central charge of the bulk theory, while the constant encodes the logarithm of the overlap between the boundary state and the vacuum. Comparing equation (C.3) and (C.4) it is possible to fix the free energy contribution from operators of dimension $\widehat{\Delta} = 2$ to be proportional to the central charge

$$c_1 = \sum_{\widehat{\Delta} = 2} \lambda_{T \widehat{\mathcal{O}}} \widehat{b}_{\widehat{\mathcal{O}}} = -\frac{\pi c}{3}. \tag{C.5}$$

The above constant corresponds to the thermal one-point functions of the stress-energy tensor in $d = 2$ (see equation B.7 in [20] or [103]) and it is therefore natural to associate this contribution to the projection of the bulk stress-energy tensor onto the defect. In fact, thermal one-point functions on the cylinder can only be associated with the vacuum module, made by $\{\mathbb{1}, T, T T, \dots\}$. Therefore only one operator contributes to the sum in equation (C.5).

## D  Stress-energy tensor and relevant operators on the defect

In this appendix, we show that, even if in general a one-dimensional defect theory can contain relevant operators in the spectrum, those do not appear in the bulk to defect OPE of the bulk stress-energy tensor, and therefore the zero-temperature limit of the entropy density derived in Section 4 is convergent. We focus on scalars since, by unitarity, if such a relevant operator appears it has to be a scalar. We want to argue that the only scalar operator that can appear in the bulk-to-defect OPE of the stress-energy tensor is the identity. We start from the bulk to defect OPE term corresponding to a scalar operator

$$T^{ij}(\tau, \vec{x}) \mathcal{P} \supset |\vec{x}|^{\widehat{\Delta} - d} \left( \delta^{ij} - \frac{d}{2} \frac{x^i x^j}{x^2} \right) \lambda_{T \widehat{\mathcal{O}}} \widehat{\mathcal{O}}(\tau). \tag{D.1}$$

The conservation of the stress-energy tensor implies that $\partial_i T^{ij}(\tau, \vec{x}) = 0$ since the derivative in the time direction produces conformal descendents which have zero one-point functions because of time translation invariance. The conservation of the stress-energy tensor implies

$$\widehat{\Delta} \lambda_{T \widehat{\mathcal{O}}} \widehat{b}_{\widehat{\mathcal{O}}} = 0, \tag{D.2}$$

which is solved either by $\widehat{\Delta} = 0$ or $\lambda_{T \widehat{\mathcal{O}}} \widehat{b}_{\widehat{\mathcal{O}}} = 0$, implying that the only scalar operator contributing to the stress-energy tensor bulk to defect OPE is the defect identity operator $\widehat{\mathbb{1}}$.

In conclusion, if the defect theory contains a relevant scalar operator this will not appear in the bulk to defect OPE of the bulk stress-energy tensor.

Let us comment that the above equation is correct because in the setup we are considering spatial translations are preserved. If we considered a similar setup on the geometry $S^1_\beta \times S^{d-1}_R$ the conservation of the stress-energy tensor would be modified and the argument presented in this appendix might not hold anymore.

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
