# Peer review of "Conformal line defects at finite temperature"

_SciPost Physics, doi:SciPost Phys. 18, 018 (2025)_

## Round 1 · Referee Report · Anonymous (Referee 1) · 2024-9-12

Report

This paper studies conformal field theories at finite temperature in the presence of a defect. The authors study one- and two-point functions. They establish sum rules and set up a bootstrap problem. The input for their approach are the T=0 in data and thermal one-point functions. They check their approach in a free field theory and make predictions for the O(n) model. The paper is very well written and new. It contains interesting material that is of interest and if important for the future study of CFTs. I recommend this paper for publication.

Recommendation

Publish (easily meets expectations and criteria for this Journal; among top 50%)

---

## Round 1 · Referee Report · Anonymous (Referee 2) · 2024-11-22

Report

This manuscript studies conformal field theory (CFT) at finite temperature with a conformal line defect extended along the thermal circle. The authors discuss how the preserved symmetries constrain simple correlation functions and give rise to new data at finite temperature. The authors then derive an inversion formula for 1-pt functions, sum rules, and a relation between thermodynamic quantities and thermal defect CFT data, before illustrating many of these ideas in two concrete examples.

The paper makes a significant contribution to the study of defects in CFT, and is likely to become a standard reference in this subject. The work is original, novel and, to the best of my knowledge, correct. The manuscript is well-structured and well-written.

Requested changes

Before I recommend this paper for publication, I would like the authors to address a few minor comments and questions:

1-Could the authors check if their powers of z in eqs. (2.16) and (2.17) should be -\Delta-1? Also, a factor of 2\pi i from the Mellin (rather than Laplace) transform seems to have gone missing.

2-Below eq. (3.5) the authors claim that only primary operators contribute in that equation. Could the authors clarify if they mean bulk or defect primaries?

3-The presentation in section 4.2 could benefit from some further clarifications, especially the discussion around eqs. (4.18) to (4.20).

4-The vanishing of the entropy at T=0 in eq. (5.19) should be argued more carefully.

5-The second term on the RHS of eq. (5.36) should read 1/2 \sigma(\phi_i\phi_i).

6-Could the authors check the subscripts in their expression for c_0 in the paragraph following eq. (5.40), and in (5.42)? I believe \sigma -> \phi_1 there. Also, the bulk-to-defect couplings in the last paragraph of sec. 5 should be normalised by c_0 if I’m not mistaken.

7-The authors may wish to add some recent references on p.32 about defect fusion at T=0, e.g. 2102.00718, 2202.03471, 2304.10239.

8-In appendix A, could the authors comment on broken Ward identities with disorder-type defects?

Recommendation

Ask for minor revision

---

## Round 2 · Author Response

to the manuscript.

---

## Round 2 · List of Changes

Referee #2: 1. We corrected the typos; 2. Time translations set to zero the one-point functions of defect descendant operators, while in general the one-point functions of conformal descendants are non-vanishing for bulk operators. However, in the limit in which the two external operators are placed at zero spatial distance (equivalently they are at the same distance from the defect), all the descendants are built by acting with time derivatives and their one-point functions are therefore zero. We added the footnote 12 with this comment; 3. We added clarifications in Section 4.2; 4. We added the footnote 19 explaining the vanishing of the zero T entropy in the free scalar theory case; 5. We corrected the typo; 6. We corrected the typos; 7. We added all the references suggested by the referee; 8. The expression (A.1) is not the most general definition of conformal defect (one exception is for instance the case of disorder-type defects). Therefore only the broken Ward identities which are not strictly dependent on (A.1) holds in more general cases (for instance eq. A.9). We added a comment in the manuscript at the end of the Appendix A.

---

## Editorial Decision

published